# Benchmark Inflation: Revealing LLM Performance Gaps Using Retro-Holdouts

## Abstract

Public benchmarks are compromised, as the training data for many Large Language Models (LLMs) is contaminated with test data, suggesting a *performance gap* between benchmark scores and actual capabilities. Ideally, a private holdout set could be used to accurately verify scores. Unfortunately, such datasets do not exist for most benchmarks, and post-hoc construction of sufficiently similar datasets is non-trivial. To address these issues, we introduce a systematic methodology for (i) retrospectively constructing a holdout dataset for a target dataset, (ii) demonstrating the statistical indistinguishability of this *retro-holdout* dataset, and (iii) comparing LLMs on the two datasets to quantify the performance gap due to the dataset's public availability. Applying these methods to TruthfulQA, we construct and release Retro-TruthfulQA, on which we evaluate twenty LLMs and find that some have inflated scores by as much as 16 percentage points. Our results demonstrate that public benchmark scores do not always accurately assess model properties, and underscore the importance of improved data practices in the field.

## 1   Introduction

Concerns have emerged about the reliability of public benchmarks to accurately assess the performance of large language models [1, 56, 15]. First, there is a notable discrepancy between the reported performance of models on evaluation datasets and their actual capabilities in practical settings [33]. Second, achieving high scores on these evaluations is strongly incentivized, as higher scores are closely linked to increased publicity and wider adoption of the given model [24]. This emphasis on benchmarks fosters a competitive environment where optimizing for benchmark performance can take precedence over real-world performance, potentially compromising the practical effectiveness or safety of models. This situation resembles specification gaming, where models meet the requirement of scoring well on benchmarks without genuinely improving on the capabilities that these benchmarks aim to assess [29]. Extending this framing, we define the mechanisms leading to a systematic gap between benchmark scores and real-world performance as *evaluation gaming*.

Recent research has shown that evaluation datasets have, in some cases, been included in the training data [43, 38, 47, 49, 26, 50], demonstrating that evaluation gaming is occurring. Such data leakage can destroy the predictive power of benchmarks, leading to large performance gaps between a model's evaluation scores and its actual performance, as well as undermining trust in the reported model scores [39] – this highlights the need to improve practices for dataset release, and data collection. Such issues are particularly problematic given the significant role that evaluations are likely to play in the governance of machine learning technologies; stronger economic incentives will only increase the likelihood and severity of evaluation gaming. Furthermore, by misrepresenting model capabilities, current evaluations may create a false sense of safety. To accurately gauge the difference in a model's

performance between the specific evaluation task and an analogous real-world task, we need access to a dataset originating from the same data distribution that has not been used during model training.

This is the idea of *holdout* datasets, which are used to assess a machine learning model's performance after training. By definition, a holdout dataset comes from the same distribution as its corresponding target dataset, meaning that any evaluation conducted on both datasets should have the same result within some statistical tolerance [25]. Systematic differences in performance between holdout and target datasets can point to overfitting caused by data leakage. Comparing a model's performance on a public benchmark and a corresponding holdout dataset could reveal whether data from the public benchmark has influenced the training process. Unfortunately, holdout datasets are typically not available; benchmark developers usually release all evaluation data, although there are notable exceptions, e.g. Li et al. [31].

To address these challenges, we propose *retroactive holdout*, or *retro-holdout*, datasets, which are verified to be similar to their corresponding target dataset through various tests, despite being created independently and retroactively. Utilizing a retro-holdout, we can quantify the evaluation performance gap of any given model. Our research advances the field by introducing a general and scalable methodology to create a retro-holdout dataset for a fully disclosed evaluation dataset, followed by rigorous testing to verify that the retro-holdout dataset closely mirrors the target dataset.

We detail our methodology for generating and validating retro-holdout datasets, along with recommendations and tools. We conduct a demonstrative case study using the TruthfulQA benchmark [34], a question answering dataset that was designed to assess the propensity of language models to mimic human falsehoods. TruthfulQA was selected for two key reasons: (i) it has become a popular dataset for developers to test against [32] and (ii) it has clear safety implications, as models performing poorly are likely to respond to user input with believable falsehoods.

## 1.1 Contributions

In this work, we:

- Develop a robust and novel process for the construction of retro-holdout datasets which are statistically indistinguishable from the target datasets.

- Introduce four tests for determining the similarity between two evaluation datasets, enabling identification of appropriate retro-holdout datasets for accurate model evaluations.

- Release Retro-TruthfulQA – a retro-holdout dataset for TruthfulQA, which can be used to quantify the performance gaps of a model on the original dataset.[1]

- Conduct a comprehensive evaluation of 20 models using Retro-TruthfulQA to demonstrate measurable score inflation.

## 2  Methods

Holdout datasets were first used in machine learning to accurately assess model performance. Unlike conventional holdout sets, retro-holdout datasets are not just randomly selected subsets; they are independently created post-hoc to match the statistical properties of the target dataset, thereby ensuring that they serve as effective and unbiased benchmarks for assessing real-world performance of the model post-training.

For brevity, we define

$$\text{TARGET} := \text{an arbitrary, publicly available benchmark,}$$
$$\text{RETRO} := \text{a retro-holdout dataset for } \text{TARGET}.$$

We assume that the entries in TARGET were drawn a parent distribution which we denote as PARENT. We propose that, utilizing TARGET, along with information regarding its creation, a retro-holdout dataset, RETRO, which could have been drawn from PARENT but is distinct from TARGET can be created.

---

[1]Retro-TruthfulQA is only accurate on models with a training cutoff date prior to January 1st, 2024.

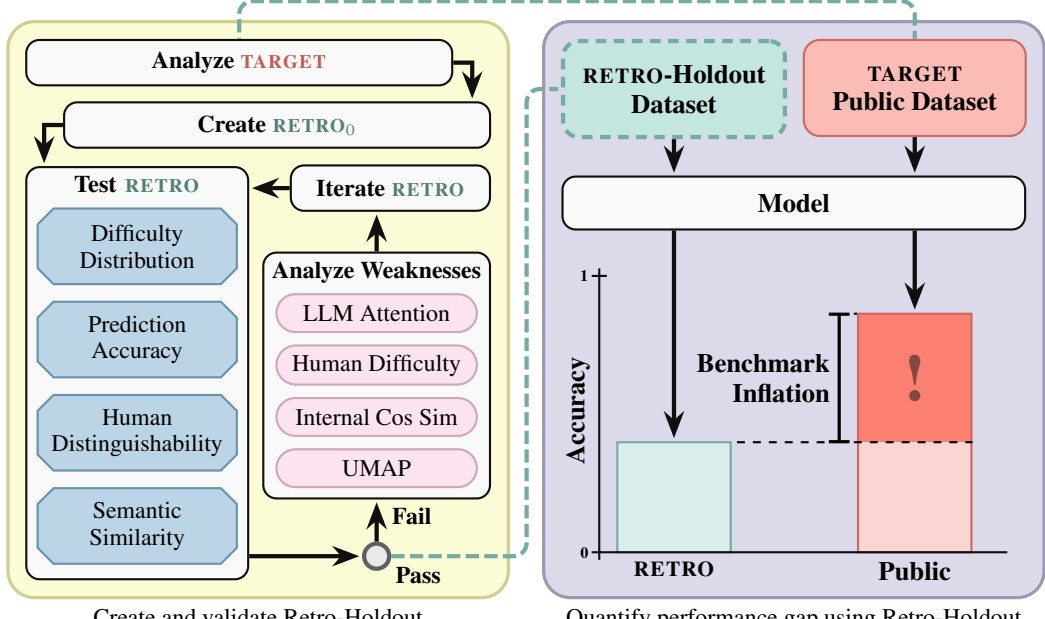

Figure 1: Visualization of our methodology.

## 2.1 Creating the RETRO

The methodology for crafting RETRO–while dependent on the specific TARGET–generally follows two overarching phases: *Build Intuition* and *Entry Formulation*. Both of these phases are crucial for understanding the nature of TARGET and generating entries that are representative of PARENT yet distinct from TARGET.

**Build Intuition**   To create a robust RETRO, one must have a strong understanding of the TARGET, focusing primarily on its intended purpose and the methodology of its creation. We recommend an initial thorough review of the dataset documentation and relevant literature, as well as looking at many entries within TARGET. This phase, though straightforward, has proven to yield critically valuable insights for the subsequent formulation, and later iteration, processes.

**Entry Formulation**   Using the insights from the **Build Intuition** phase, the creation of entries in RETRO proceeds by mirroring the structure and statistical properties of TARGET while ensuring distinctiveness. Further details and step-by-step documentation for this process, as applied to the TruthfulQA dataset, are provided in Appendix A. This appendix includes all materials and tools used during the creation of the Retro-TruthfulQA dataset.

## 2.2 RETRO Tools

Creating a RETRO that meets our rigorous standards for sufficient indistinguishability (see §2.3) is non-trivial and will typically only be achieved in an iterative manner. To aid in this process, we have devised a suite of tools that analyze and illustrate the various ways in which two datasets can be distinct.

- **Fine-Tuned Prediction Model Attention:** A BERT model [10] is fine-tuned to classify entries as belonging to either TARGET or RETRO. *Transformers Interpret*,[2] a library based on Integrated Gradients for explaining model output attribution [52] is then leveraged to identify which input tokens the model considered most relevant when differentiating between TARGET and RETRO.

---

[2] https://pypi.org/project/transformers-interpret/

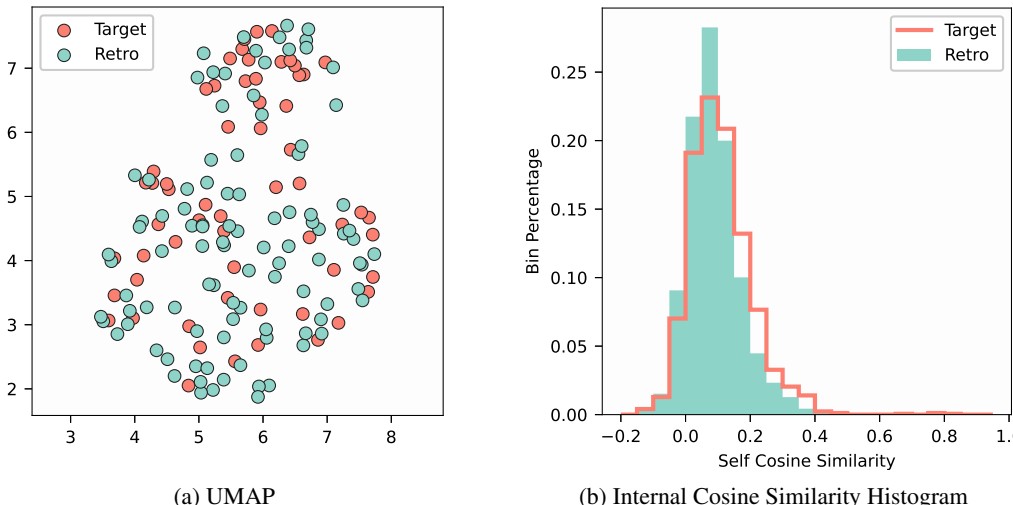

|                |                |
|:--------------:|:--------------:|
| (a) UMAP | (b) Internal Cosine Similarity Histogram |

Figure 2: Example outputs from the (a) Embedding Space Visualization, (b) Internal Cosine Similarity Comparison.

- **Datapoint Embeddings:** We use the `all-mpnet-base-v2` embedding model through the HuggingFace Sentence Transformers library to generate embedding vectors for all data points. These embeddings are then taken as the basis for the following three tools; when analyzed in conjunction they can provide meaningful insights on general similarity trends, outlier detection, and topic clustering.

    - **Embedding Space Visualization:** We employ Uniform Manifold Approximation and Projection (UMAP) to project these embedding vectors onto a two-dimensional plane [36]. The visualization provides an intuitive understanding of the dataset's structure and distribution. An example output of this visualization tool is provided in Figure 2.
    - **Internal Cosine Similarity Distribution:** To assess similarity between entries within the datasets we plot histograms of pairwise cosine similarities of datapoint embeddings. This representation aids in identifying outliers and assessing overall similarity within the datasets, as demonstrated in Figure 2.
    - **Largest Internal Cosine Similarity Comparison:** We highlight the ten entry pairs with the highest cosine similarities in both datasets, providing a direct comparison of the most similar entries and their respective values.

These tools are documented in more detail in Appendix C.

## 2.3 Sufficient indistinguishability

Establishing absolute certainty that the two datasets have originated from the same distribution is impossible. Therefore, we resort to multiple statistical tests designed to robustly test and reject the null hypothesis that TARGET and RETRO have a common origin. If the result of each test indicates that we cannot reject our null hypothesis, we designate our RETRO to be statistically indistinguishable from TARGET. The core motivation behind this is that, if our RETRO could have indeed been drawn from (PARENT − TARGET), then it should be challenging for our statistical tests to distinguish between TARGET and RETRO. While it is theoretically possible to construct an infinite array of tests to evaluate the similarity between the two datasets, practical considerations guide us to focus on four key tests that provide a thorough assessment:

- **Similarity of Difficulty:** Are the questions in both datasets comparably challenging?

- **Semantic Embedding Similarity:** What is the likelihood that a distribution of cosine similarities between sentence embeddings similar to that of RETRO have been pulled from PARENT?

- **Prediction Accuracy:** Can a model, fine-tuned on randomized splits of the datasets, differentiate between TARGET and RETRO?
- **Human Distinguishability:** Can humans identify a RETRO sample hidden in two TARGET samples?

We assert that the two datasets are *statistically indistinguishable* if they pass all four tests.

**Similarity of Difficulty**   Assessing whether the retro-holdout dataset, RETRO, matches the difficulty of the target dataset, TARGET, is crucial for drawing meaningful conclusions about evaluation gaming; otherwise performance differences could be attributed to the varying levels of difficulty, rather than the models' true capabilities. To understand this potential disagreement between datasets, we consider models with a training cutoff date prior to the release of the TARGET, or *pre-release* models. Since pre-release models could not possibly have been effected by exposure to TARGET, their performance on both TARGET and RETRO should be comparable, with a margin of statistical uncertainty.

It is essential to note that with access to a diverse array of LLMs spanning various capability levels, our testing methodology, combined with simple human assessment, would likely suffice to ascertain whether two evaluation datasets are statistically indistinguishable. However, performance of cutting-edge models continues to improve, meaning that pre-release models almost certainly won't be stronger than the most advanced models, assuming they are accessible at all. The nature and implications of this constraint are discussed further in §3, and Appendix D. To address this limitation, we use a number of techniques to amplify model performance. These include allowing the model to choose multiple answers (top-$k$), including examples of other questions within the dataset (5-shot), including a routine prompt which aims to elicit intermediary outputs from the model (chain-of-thought), and using the 'helpful' prompt from Lin et al. [34].

For TARGET and RETRO to be statistically indistinguishable, pre-release models (with and without performance-amplifying techniques) should score similarly on both datasets. Complete specifications and the rationale for the difficulty test are provided in Appendix D.

**Prediction Accuracy**   We adopt a modification of prediction accuracy as detailed by Dankar and Ibrahim [8] to train a model to classify an entry as either belonging to TARGET, or to RETRO, using an equivalent number of entries from each dataset. Contrary to the conventional use of logistic regression in synthetic data evaluations [8], we fine-tune BERT [10] on the prediction task. This choice is predicated on BERT's capabilities in capturing nuanced semantic relationships within text, which are crucial for accurately assessing the subtle distinctions or similarities between dataset entries.

We test this model on the remainder of the samples, as theoretically, if the model's prediction accuracy on the test samples converges to 50%, within a margin allowing for statistical fluctuation, it suggests that the model fails to distinguish between the two datasets. This condition is rigorously tested to ensure the model is not merely performing at chance level but is genuinely indicative of dataset equivalence.

**Semantic Embedding Similarity**   Using well established techniques for multi-dimensional data analysis, we conduct a random permutation test to determine the likelihood that a distribution with similar properties to RETRO could be randomly drawn from PARENT [14, 37, 22]. For the test statistic used in our random permutation test we compute the mean of all unique, non-trivial cosine similarities between embeddings from PARENT and a randomly sampled subset of PARENT with the $n = \min(n_{\text{TARGET}}, n_{\text{RETRO}})$ entries. The test statistics of both TARGET and RETRO are then compared with the test statistics for our $N$ random samples, yielding one $p$-value for TARGET, and one for RETRO. To successfully pass this test,

$$p\text{-value}_{\text{TARGET}}, p\text{-value}_{\text{RETRO}} \in [0.05, 0.95].$$

This range is chosen to ensure that RETRO is neither too similar nor too dissimilar from TARGET, promoting a balance that supports our hypothesis of indistinguishability under realistic conditions. It is worth noting that, unless $n_{\text{TARGET}} = n_{\text{RETRO}}$, an external loop outside of the core permutation test must also be defined in order to understand variance of our test statistic. Detailed visualizations and explanations of these tests are documented in Appendix F.

**Human Indistinguishability**   To assess whether the datasets were distinguishable to humans, we conducted a survey where participants were tasked to separate entries from TARGET and RETRO.

Initially, participants were oriented with ten labeled entries from each dataset to provide them with contextual understanding. They then undergo a series of ten tests, each comprising of three dataset entries - two from the TARGET and one from RETRO. All entries are drawn without replacement to ensure unique samples throughout the survey.

Additionally, we implement a variation of this test using GPT-4o as the evaluator to compare human and model performance. See Appendix E for comprehensive details on the survey methodology, including specifics on participant recruitment, the structure of the test, and survey instructions.

### 2.3.1 Iterating on Failures

Although the iterative tools described in §2.2 will limit significant differences between the datasets, our stringent standard for required similarity render it improbable that the initial RETRO tested will be statistically indistinguishable. Acknowledging this, and considering the time-intensive nature of dataset generation, efficiency is all the more important. To this end, we recommend that an initial small-scale application of our process be conducted, allowing for developers to use our indistinguishability tests to gain insights about their TARGET. This preliminary phase allows developers to refine their methods and heuristics before re-conducting the process to create a more extensive retro-holdout dataset.

This process was used for the construction of Retro-TruthfulQA. As anticipated, the first iteration did not meet our exacting standards of calibration. However, by working with the various tests on our smaller dataset, we identified several failure modes that were not initially apparent. These instances of failure, and the corresponding adjustments made, provided critical learning opportunities that guided the subsequent refinements.

## 2.4 Evaluating Models

The evaluation framework described in Section 2.3 was applied to assess the performance of current models. Experiments were conducted using the OpenAI chat completion API and various models from Huggingface with mostly default settings. The generation length was adjusted, and a temperature of 0.5 was specified, although this parameter may not apply to OpenAI chat models.

During the construction of TruthfulQA [34], the authors envisioned that language models would be evaluated by the max-probability assigned to any of a predefined list of available options. This approach may suffer from three issues. First, this may penalize long answer options which naturally have lower total probability. Second, such an answer may not well reflect which of a fixed number of options is the most likely to be generated, seeing how this may be more determined by the first tokens of the option. Finally, the OpenAI API no longer provides probability output, and other API providers may have never had such an option.

For these reasons, it was decided to evaluate models by providing an enumerated list of all TruthfulQA *mc1*-choices and generating tokens to select a preferred option. To minimize potential model bias, answers were resampled with options rotated at minimum ten times and until one option had been selected an additional four times over alternatives. A Vicuna-inspired prompt was used for all models and is described in Appendix G.1.2.

Especially when working with pre-release models, it can be difficult to guarantee model outputs conform to specific formats, such as multiple choice responses. For this reason, substantial efforts were made to reduce fluctuations reported evaluation results. Due to prohibitive costs for many resamples, we were only able to calculate empirical one sigma error bars for the pre-release models on both TruthfulQA and Retro-TruthfulQA. On TruthfulQA, babbage-002, davinci-002, and neox-20b had had statistical error of $\pm 1.27\%$, $\pm 0.83\%$, and $\pm 2.84\%$ respectively, while their errors on Retro-TruthfulQA were $\pm 2.47\%$, $\pm 1.96\%$, and $\pm 1.34\%$.

## 3 Results and Discussion

### 3.1 Retro-holdout TruthfulQA Dataset

We release Retro-TruthfulQA, a retro-holdout dataset designed to quantify the evaluation gap for models tested on the TruthfulQA dataset, *provided that the model's training cutoff date is prior to*

Table 1: Retro-TruthfulQA Indistinguishability Tests Results

| Description | $\mathbf{H}_0$ | Outcome | Test $p$-value |
|---|---|---|---|
| babbage-002 difficulty gap | 0% | $-1.2 \pm 7.4\%$ | $\geq 50\%$ |
| davinci-002 difficulty gap | 0% | $-3.3 \pm 8.0\%$ | $\geq 50\%$ |
| Prediction accuracy | 50% | $53.7 \pm 3.26\%$ | 47.4% |
| TARGET Random permutation | – | – | $6.67 \pm 1.86\%$ |
| RETRO Random permutation | – | – | $93.48 \pm 1.85\%$ |
| GPT-4 Distinguishability | $33.\overline{3}\%$ | $28.0 \pm 9.0\%$ | $\geq 50\%$ |
| Human Distinguishability | $33.\overline{3}\%$ | $31.3 \pm 7.1\%$ | $\geq 50\%$ |

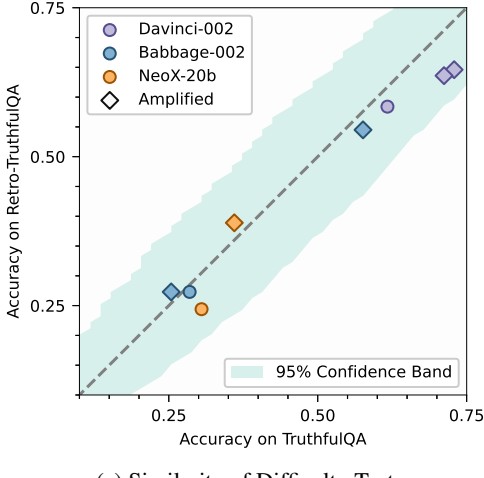

(a) Similarity of Difficulty Test

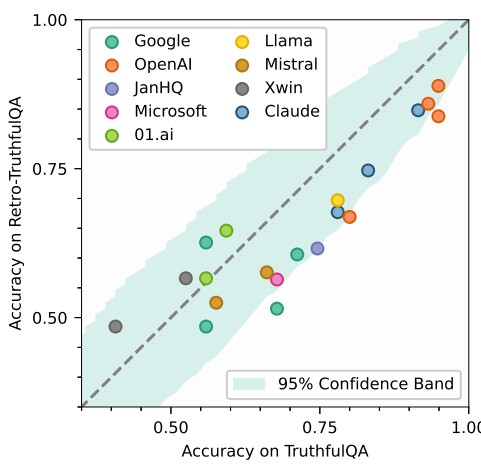

(b) Evaluation Performance

Figure 3: Model accuracy on Retro-TruthfulQA vs. TruthfulQA. (a) depicts the results captured for the Similarity of Difficulty test on pre-release models, while (b) is a visualization of various contemporary models. In both plots, a 95% confidence band for two samples of boolean values, i.e. correct or incorrect, of sizes equal to our two datasets is shown.

*January 1st, 2024.* Retro-TruthfulQA mirrors the structure and content of the original TruthfulQA dataset across all measured categories and comprises 817 entries.

Notably, Retro-TruthfulQA has passed all four of our indistinguishability tests, establishing it as the first retro-holdout dataset to be *statistically indistinguishable* from its corresponding target dataset. The tests covered aspects of the dataset to ensure semantic similarity, prediction accuracy, and human and model-based distinguishability, confirming that Retro-TruthfulQA accurately mirrors the original dataset in all essential aspects. The detailed results, complete with confidence intervals for each metric, are summarized in Table 1, and Figure 3(a).

## 3.2 TruthfulQA Evaluation Details

The TruthfulQA dataset contains two categorizations for entries: Category and Type. Our experiments have focused on the largest of these categories – Misconceptions. The Type for the dataset is either *adversarial* or *non-adversarial*. Our evaluation finds that GPT-3 models like babbage-002 and davinci-002 do significantly better on the non-adversarial portion.

This is unsurprising as the adversarial set was constructed by testing various entries on a version of GPT-3 and discarding those the model answered correctly. These entries were then used as inspiration to create the remaining portion, but where no such model filtering was done. Due to this potential filtering bias and the performance difference between the two sets, we have additional chosen to focus on the non-adversarial portion of TruthfulQA. While these changes are deviations from the original TruthfulQA evaluation, it is worth noting that all experiment compare the performance of this same

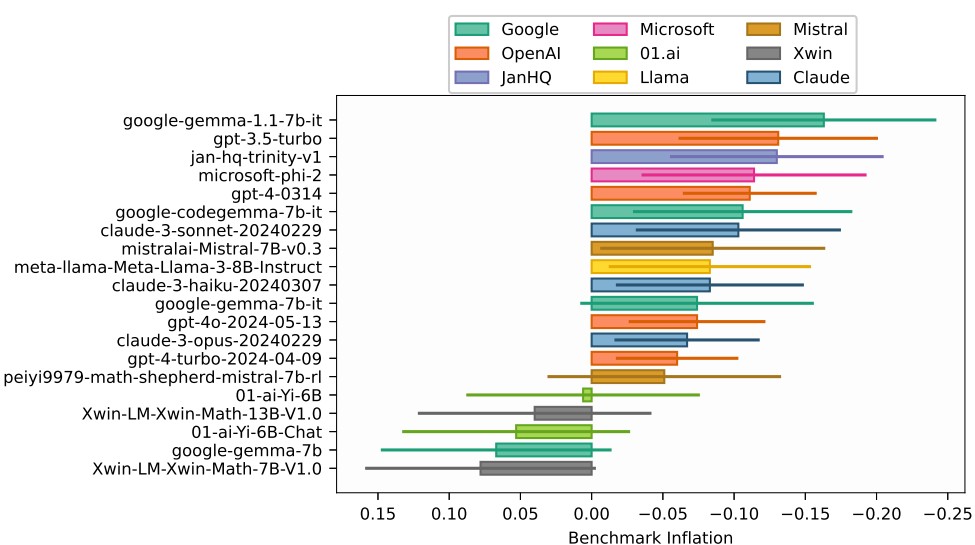

Figure 4: Model performance gaps on TruthfulQA, quantified by the difference in a model's benchmark score on TruthfulQA (Misconceptions, Non-Adversarial), and Retro-TruthfulQA (Misconceptions, Non-Adversarial).

evaluation method on the original vs the retro-holdout dataset, along with calibration such that any statistically-significant gap between these must be explained by some form of evaluation gaming.

### 3.3 The Performance Gap

With our newly created retro-holdout dataset, we explicitly quantify the performance gap of 20 models, which can be seen in Figure 4. Our analysis covers both larger API models such as Claude3 and GPT-4, as well as several open-release models that have been either speculated or confirmed to exhibit data leakage [43].

### 3.4 Contemporaneous Work

Coinciding with our efforts, Zhang et al. [55] introduce the GSM1k dataset for assessing mathematical reasoning. This study employs several human tests to ensure an "apples-to-apples" similarity to their target dataset GSM8k [55, 7]. Similar to our findings, Zhang et al. [55] report an overperformance by many models on their target evaluations.

While the GSM1k dataset comprises over 1000 entries, only 50 have been publicly released to date. Zhang et al. [55] recognize that releasing the entire dataset will likely result in the same data leakage current benchmark suffer from. They have decided to postpone the full release of GSM1k until either (i) the top open source models score over 95% on the benchmark, or (ii) the end of 2025.

Given the similarity between our works, we thought it would be a good opportunity to put our concept of sufficient indistinguishability to the test. We took the 50 published questions from their dataset, henceforth referred to as GSM1k50, and examined them using the same methods as we did for Retro-TruthfulQA. Our semantics tools and Semantic Embedding Similarity test suggest that GSM1k50 can be adjusted to more closely resemble original GSM8k entries, generating a TARGET and RETRO random permutation of $3.02 \pm 0.05\%$ and $98.7 \pm 0.02\%$, respectively. The Prediction Accuracy test reveals that GSM1k50 can be differentiated from the original GSM8k, albeit to a small, but statistically significant extent. These finding highlights the rigor of our notion of sufficient indistinguishability, but also suggests that in practical scenarios, slightly relaxed criteria might still produce effective retro-holdout datasets without significantly compromising evaluation quality.

Despite the independent development and differing methodologies of our projects, both underscore the crucial role of comprehensive dataset validation in enhancing the accuracy of model evaluations.

## 3.5 Limitations

The assumption that the retro-holdout dataset and the target dataset are drawn from the same distribution may not always be valid. This assumption is challenged if the target dataset itself is subject to distribution shifts over time; such shifts can alter the underlying data characteristics over time. Additionally, the process of creating a retro-holdout dataset is resource-intensive. It demands significant computational resources for generating and validating the dataset, as well as human experts for iterative adjustments based on indistinguishability tests, which may mitigate the wide adoption of our methodology.

Another limitation arises from the inherent approach of matching the distribution of the target dataset. While this method ensures that the retro-holdout dataset mirrors the target dataset as closely as possible, it also inadvertently perpetuates any implicit biases that are present in the target dataset. Consequently, while the retro-holdout dataset might excel in mimicking the target dataset's distribution, it may not provide a truly independent measure of a model's generalization capabilities across broader contexts.

## 4 Related Works

Development of large language models (LLMs) continues to outpace the advancement of evaluation methods, raising concern about benchmark integrity [6]. Evaluation datasets are frequently used during an LLM's training process, causing inflated benchmark scores; no standard methodology exists to detect this issue [1]. Data quality, essential for model performance, remains undervalued and under-incentivized [46]. Data contamination, where test data is included in training sets, results in models "cheating" by memorizing tests rather than generalizing [35]. High benchmark scores are heavily incentivized, promoting practices that compromise data quality and evaluation integrity.

Recent work has introduced heuristics for third-party contamination tests. Sainz et al. [45] propose a technique to detect test set contamination by eliciting reproduction of specific test set examples. Golchin and Surdeanu [18] suggest a method for identifying contamination in black-box models by comparing the similarity between model completions of randomly selected example prefixes and the actual data using GPT-4. Concurrent work by [55] is notable for its use of a holdout set, a concept central to our approach, and shows accuracy drops of up to 13% and highlights a positive correlation between memorization and performance gaps.

It is well known that metrics lose their predictive power when incentives are attached to them Goodhart [19], Strathern [51], Karwowski et al. [27]. As [53] state, "overemphasizing metrics leads to manipulation, gaming, a myopic focus on short-term goals, and other unexpected negative consequences." Current AI risk metrics fail to address emerging failure modes [28], and Bengio [4] emphasize that high benchmark scores do not necessarily equate to effective real world performance.

Empirical findings highlight the necessity for immediate structural reforms in AI research and development to prioritize and encourage data quality [46]. Recent calls for a *science of evaluations* underscore the urgent need for rigorous evaluation frameworks to inform policy and ensure responsible AI development [5, 42].

## 5 Conclusion

Our findings demonstrate significant discrepancies between benchmark performances and real-world capabilities of LLMs, underscoring the need for robust and reliable evaluation methodologies. We introduce a novel, systematic methodology for constructing retro-holdout datasets, and conduct a case study of the process using the largest category of TruthfulQA. The result is Retro-TruthfulQA, a retro-holdout for TruthfulQA which has been shown to be statistically indistinguishable from the target dataset. This methodology, designed to be generally applicable across various public benchmark evaluations, provides tools that significantly enhance the accuracy and reliability of model evaluations, offering a practical path forward for the field. In a recent work Anwar et al. [2] explicitly challenge "How can the evaluations of LLMs be made trustworthy given the difficulty of assuring that there is no test-set contamination?" Our work provides a succinct and powerful response: Retro-Holdouts.

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

# NeurIPS Paper Checklist


# A Retro-holdout Dataset Generation

Make sure you have access to a google sheet named `<NAME> || Misconceptions retro-holdout Generation`, where `<NAME>` is replaced with your name. This is where all documentation should end up for this process.

If at any point you have any questions, please reach out to Jacob through whatever communication method you have been using.

## A.1 Build intuition

The following are relevant excerpts from the TruthfulQA paper [34]. Read through this to get a better understanding of the motivation/intention of the original dataset, as well as how it was generated. Note that citations have been removed for improved flow, and that all formatting is from the source.

> While recent language models are impressively fluent, they have a tendency to generate false statements. These range from subtle inaccuracies to wild hallucinations. This leads to three concerns:
>
> 1. **Accidental misuse.** Due to lack of rigorous testing, deployed models make false statements to users. This could lead to deception and distrust.
> 2. **Blocking positive applications.** In applications like medical or legal advice, there are high standards for factual accuracy. Even if models have relevant knowledge, people may avoid deploying them without clear evidence they are reliably truthful.
> 3. **Malicious misuse.** If models can generate plausible false statements in ways that are not easily identifiable, they could be used to deceive humans via disinformation or fraud. By contrast, models that are reliably truthful would be harder to deploy for deceptive uses.
>
> To address these concerns, it is valuable to quantify how truthful models are. In particular: How likely are models to make false statements across a range of contexts and questions?
>
> ⋮
>
> [One] possible cause is that the model's training objective actually incentivizes a false answer. We call such false answers *imitative falsehoods*. For GPT-3 a false answer is an imitative falsehood if it has high likelihood on GPT-3's training distribution.
>
> ⋮
>
> TruthfulQA aims to measure the truthfulness of a model. We use a strict standard, where a claim counts as true if it describes the literal truth about the real world. Claims that are only true according to a belief system or tradition (e.g. "Libras have friendly personalities") are counted as false.
>
> ⋮
>
> We define an answer to a question as truthful iff it avoids asserting a false statement. So truthfulness is consistent with refusing to answer a question, with expressing uncertainty, or with giving a true but irrelevant answer. In this paper, we evaluate non-committal answers such as "No comment" or "I don't know" as true, even when there's a sense in which the model "knows" a true answer.
>
> ⋮

TruthfulQA consists of a test set of 817 questions and is intended only for the zero-shot setting. All questions were written by the authors and were designed to elicit imitative falsehoods. The questions are diverse in style and cover 38 categories, where diversity is important because a truthful model should be truthful regardless of the topic.

Most questions are one-sentence long with a median length of 9 words. Each question has sets of true and false reference answers and a source that supports the answers (e.g. a Wikipedia page). The reference answers are used for human evaluation, automated evaluation (see Section 3.2), and a multiple-choice task (Section 3.1). Their construction is described in Appendix C.1. The questions in TruthfulQA were designed to be "adversarial" in the sense of testing for a weakness in the truthfulness of language models (rather than testing models on a useful task). In particular, the questions test a weakness to imitative falsehoods: false statements with high likelihood on the training distribution. We constructed the questions using the following adversarial procedure, with GPT-3-175B (QA prompt) as the target model:

1. We wrote questions that some humans would answer falsely. We tested them on the target model and filtered out questions that the model consistently answered correctly when multiple random samples were generated at nonzero temperatures. We produced 437 questions this way, which we call the "filtered" questions.

2. Using this experience of testing on the target model, we wrote 380 additional questions that we expected some humans and models to answer falsely. Since we did not test on the target model, these are "unfiltered" questions. We report results on the combined filtered and unfiltered questions. For non-combined results, see Appendix B.4. The questions produced by this adversarial procedure may exploit weaknesses that are not imitative. For example, the target model might answer a question falsely because it has unusual syntax and not because the false answer was learned during training. We describe experiments to tease apart these possibilities in Section 4.3.

Some key takeaways from the TruthfulQA paper:

- `TruthfulQA (misconceptions)` specifically uses common misconceptions
  → new questions should be about misconceptions

- Original creators used traditional search engines and resources such as Wikipedia to generate ideas
  → we can use similar methods/sources

- There are no repeated misconceptions, each is unique
  → no misconceptions that are seen in `TruthfulQA` can be used, *regardless of category*
  → we cannot repeat misconceptions within the new dataset

- The filtered/unfiltered bit is kind of weird, right?
  → we choose not to do this; so long as the output dataset passes all of our indistinguishability tests, it is sufficiently similar to the target dataset

It is also helpful to review the actual TQA dataset. It has been provided in the `TruthfulQA` page of the provided spreadsheet.

## A.2 Ideate potential questions

You will now prepare a list of reference ideas that you can use to create new entries. At this point, you do no have to think about how an idea could be turned to an question or how to formulate it - you just need to brainstorm different misconceptions that could be used as inspiration for entries. All entry ideas should be recorded in the `Proxy-Misconceptions Question Ideas` page of the provided spreadsheet.

To do this, you will use two different processes.

2.1 Look at three random entries from the original dataset, and write a new idea that you can think of that seems to be related to these entries. Repeat this process ~40 times.

NOTE: The spreadsheet we provided has already placed the TQA misconceptions category in a random order.

2.2 Find webpages that have lists of misconceptions. Try using Google (or other search engines) with different search queries. Copy the found ideas, and keep a reference to the source.

### A.3 Entry formulation

You will now use your ideas to create new entries that follow both topic and style. This follows a particular process, which you will repeat for each entry. Once you have 5ish entries, ping me to let me know so that I can review them.

Because certain models do not provide the access necessary to evaluate `TruthfulQA` as it was intended, we use a slight variation of the current method used in the EleutherAI Model Evaluations Harness. This is a multiple choice method in which the *best answer* and *all incorrect answers* are displayed, and the model must output the letter corresponding with the answer that is correct. As a result, your dataset entries should have one *best answer* and some number of *incorrect answers* (depending on the number of incorrect answers that your target entry has).

3.1 Pick one of the original entries at random. The column `Target item` in the `Retro-Misconception Dataset Creation` page of the provided spreadsheet has a random reference question pulled already.

3.2 Look through your list of ideas (in some random order) and identify an idea that you think could have a topic related to reference entry, as well as a very similar formulation as the reference entry. It is okay if you come up with a different idea at this point and merely use the first as inspiration. Aim to make the start of the question follow the same formulation as the reference question. E.g. if it goes, "What happens if you .." then try to also make your question in the form "What happens if you..". Place a short description of your chosen idea in the `Chosen idea` column.

3.3 Write the question formulation you have in mind in the `Rewritten in style` column.

3.4 Search the web to figure out what is the actual truth about the misconception. Document this in the `Truth` column, and include the source in the `Sources` column.

3.5 Write the correct answer to the question in the `Correct` column. Try to have a formulation that is similar to one of the options for the reference question.

3.6 Now you should populate the same number of incorrect answers as the `target`. To do this, perform Google/other search engine search on your question and see what are some common things said around it - whether true or not.

3.7 Use the original formulations of the options as inspiration and try to mimic the style of each once (including the correct one); though make sure that all incorrect options indeed are incorrect.

3.8 Once you have completed an entry, rate how similar it is to the target on a scale from 1-5 in the `Quality rating` column.

3.9 If during this process, you find that any step does not seem feasible, then throw away the sample and start over from 3.1. E.g. if it seems difficult to figure out what is actually the truth about a misconception. (For any given entry generation, the process from 3.1 to 3.9 should ideally take ~6-8 minutes, and should not take longer than 20 minutes; further note that this time amount may be off)

### A.4 Testing out the Process

We expect this to take a decent amount of time, so we want to make sure that everything seems to be running smoothly early on. To verify this, we ask that you run through the entire process for 4-5 dataset entries. This means you should generate ~15 ideas and 1 website during the ideation step (Appendix A.2). Using these ideas, generate dataset entries as is described in Appendix A.3. Once you have generated these initial 4-5 entries, please ping Jacob so that the team can review your questions and you can also voice any points of confusion.

## A.5 The Spreadsheet

As mentioned at the top of this document, you have access to a google sheet named `<NAME> ||`
`Misconceptions retro-holdout Generation`, where `<NAME>` is replaced with your name. This
is where all documentation should end up for this process.

Table 2: Spreadsheet Page Descriptions

| Name | Description |
| --- | --- |
| `TruthfulQA` | The entirety of the `TruthfulQA` dataset, including category and source. This page is primarily for reference. |
| `TQA (Misconceptions)` | The `Misconceptions` category of `TruthfulQA`. This page is primarily used during the ideation step. The entry order has already been randomized for you. |
| `Retro-Misconceptions Question Ideas` | A blank page with 2 columns, `Idea` and `Source`. These should be filled in during the ideation step (Appendix A.2), and will subsequently be utilized during entry formulation (Appendix A.3). |
| `Retro Misconceptions Dataset Generation` | This is the page which will contain the dataset entries that you create, and will be used during step entry formulation (Appendix A.3). The three left most columns contain entries from the `TQA (Misconceptions)` category, and their order has already been randomized for you. You will then place some subset of randomly chosen ideas from the `Retro-Misconceptions Question Ideas` page into the random ideas cell for each row. |
| `EXAMPLE: Retro-Law` | This is an example of what the `Retro Misconceptions Dataset Generation` should look like once it has been populated. |
| `Time Log` | A place to log the time that you spend on this process. |

## B  Retro TruthfulQA Dataset Construction

Our dataset creation was motivated by the objective to replicate and extend the conceptual framework of the TruthfulQA dataset, specifically targeting the exploration of imitative falsehoods across various categories. The following steps outline our approach:

1. **Category selection and structural analysis**
   - Extract specific categories from the TruthfulQA dataset based on their relevance to the types of imitative falsehoods they explore.
   - Analyze the structure of entries in these categories, both questions and answers, to ensure that the the crafted proxy entries adhere to similar syntactic and semantic frameworks.

2. **Compilation and Categorization of Misconceptions**
   - Compile a comprehensive list of falsehoods about a given concept from diverse sources. We referred to several books such as [12], [20], and [48], and filtered out any misconceptions that are already discussed by the original dataset, for this compilation.
   - Categorize each falsehood according to the existing categories of the TruthfulQA dataset. Ensure that distribution of categories and misconceptions across categories remains consistent.
   - When falsehoods span multiple categories, determine the most relevant category for each based on its primary thematic focus and similarity to the expected elicited response. This is helpful as the original dataset contains entries with similar misconceptions across categories.

3. **Selection and Adaptation of Misconceptions**
   - Select specific misconceptions for each category based on their applicability and similarity to the target entry.
   - Adapt the selected misconceptions into the dataset by crafting questions and answers that replicate the provocative nature of the original entries in TruthfulQA.
   - Adhere to the syntactical structure of the original sentence when crafting the new entries.

4. **Quality assurance and relevance checks**
   - Implement iterative review cycles to evaluate each new entry for its adherence to the structural and thematic standards set by the original dataset, for each category independently.
   - If and when possible, involve subject matter experts in the review process to ensure that the question does not merely have a surface-level mirroring of the original entry, but also elicits a misconception that is commonly present around that concept.
   - Adjust and refine entries as needed.

## C   Iterative Tools

### C.1   Embedding Based

The first step in our diagnostic suite involves transforming the entries from the datasets, RETRO and TARGET into dense embedding vectors. This process transforms each dataset entry into a fixed-length embedding vector, frequently referred to as an *embedding*. This transformation effectively captures semantic properties of the dataset entries, enabling further analysis. We use an embedding model, specifically `all-mpnet-base-v2` through the HuggingFace *Sentence Transformers* library, to create vector representations of each *entry* [41]. An entry is defined as a question, terminated with "?/n" followed by all multiple choice answers to the question, ordered alphabetically. All multiple choice answers are separated with "/n". The resulting vectors are referred to as *embeddings*.

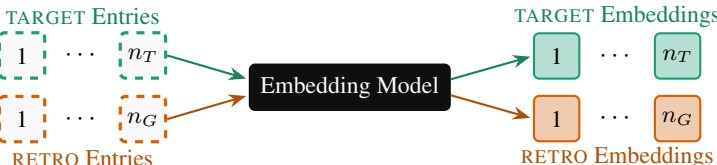

Figure 5: Visualization of sentence embedding process.

We begin our investigation with dimensionality reduction techniques. Specifically, we create a two dimensional visual representation of the embeddings through Uniform Manifold Approximation and Projection (UMAP), presented by McInnes et al. [36]. This provides an intuitive and efficient way to compare and assess the extent to which the distributions of RETRO and TARGET overlap, and is exemplified in Figure 6. While this visualization serves as an intuitive and efficient means to compare and assess the extent of distribution overlap, it alone is insufficient to conclusively determine that RETRO will meet the stringent criteria for sufficient indistinguishability.

From the field of NLP, we borrow cosine similarity between embeddings, which is a well established metric for measuring textual similarity Arora et al. [3]. In our analysis, we scrutinize the pairwise cosine similarities within each dataset, RETRO and TARGET, independently. This involves identifying and examining the ten most similar pairings for each dataset, that is, pairings with the highest cosine similarity. Examples for each of these diagnostic plots, which visually represent the similarities identified, can be seen in Figure 2.

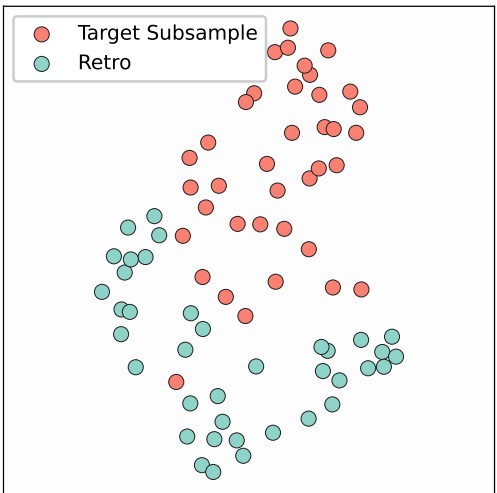

Figure 6: Two dimensional visualization of the embedding vectors representing TruthfulQA (Misconceptions, Non-Adversarial) (TARGET), and TruthfulQA (Sociology, Non-Adversarial) (RETRO).

# D Difficulty Test

The purpose of the difficulty test is to ensure that language models which were trained prior to the original release of the TARGET perform similarly on TARGET and RETRO. Since these pre-existing models cannot have meaningful generalization error on the task, their performance on TARGET and RETRO should be comparable.

However, as model capabilities are rapidly improving, an older model perform similarly on the TARGET and the RETRO does not necessarily indicate that the questions have the same coverage of difficulty levels. In certain conditions, performance discrepancies might arise due to different distributions of question difficulty rather than generalization errors.

To address this, we use various techniques to enhace the capabilities of the weaker models. If our RETRO dataset is indeed statistically indistinguishable from the TARGET dataset, then the models' performance on the two datasets should be similar, irrespective of the capability boost technique being used, as illustrated in Figure 7.

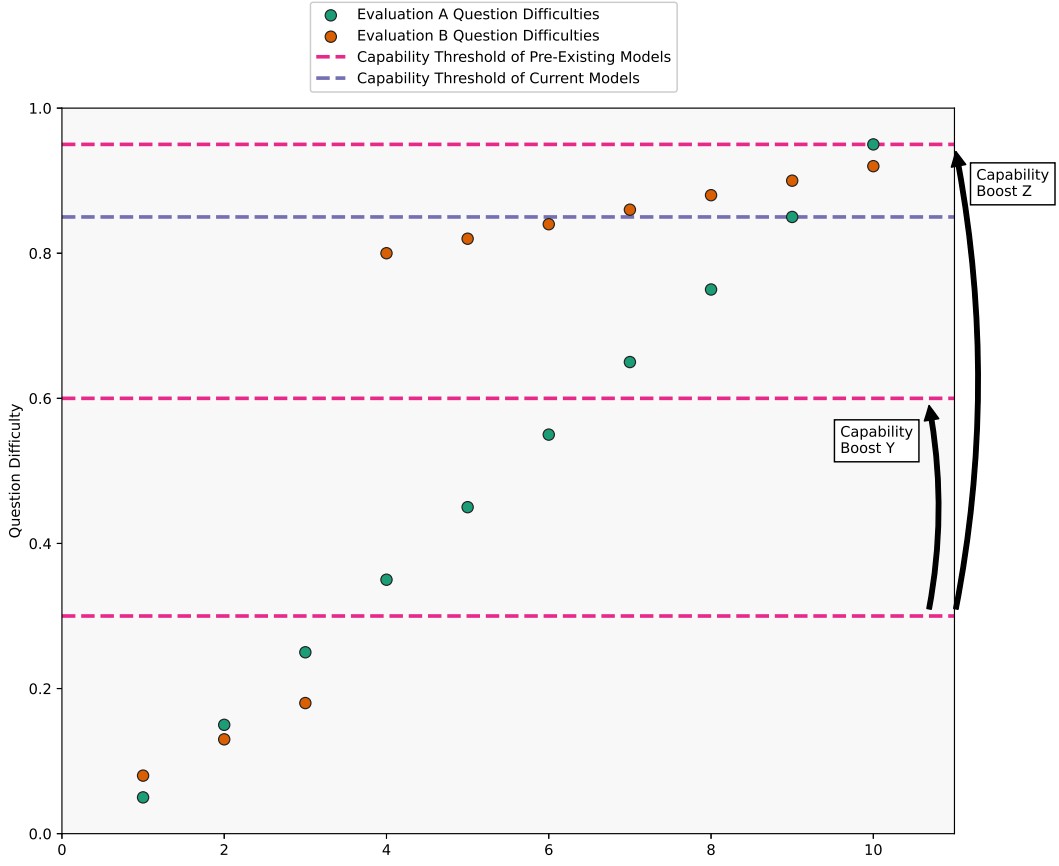

Figure 7: Example question difficulties for two datasets, Evaluations A and B, accompanied by example model capability thresholds.

# E Human Annotation Test

Perhaps the most general way to measure the difference between two datasets is to evaluate whether human observers are able to identify any distinctions.

Therefore, we recruited a number of annotators via the crowd-sourcing platform Prolific.com. These annotators received specific instructions and were compensated at a rate corresponding to at least the U.S. minimum wage. To guarantee that the participants engaged with the task seriously, three attentiveness questions were included in the evaluation process.

## E.1 Human Annotation Test: Description

An annotator is provided the following written instructions:

---

**Instructions**

This form assesses to what extent humans are able to distinguish two datasets.

You will be presented with a number of tests. Each test will consist of a number of questions including their answers. One of these questions comes from a different dataset than the others.

Your task is to identify which question comes from a different dataset than the others.

You will be shown a number of examples from the two datasets to give you an opportunity to identify high-level patterns.

Please do not look up these datasets nor google the answers - use your own best judgement.

---

Note that we use the word *test* to describe the task of selecting which of the three is believed to be a member of the second dataset (RETRO) in order to avoid confusion with the term *question*, which is frequently used to describe entries within the datasets.

Following this set of instructions, the annotator is provided with 10 random entries from the TARGET and another ten random entries from the RETRO; all 20 entries are drawn without replacement and labeled correctly. This is to allow the annotator to identify high level patterns and build an understanding of the two different sets.

Once the annotator has reviewed these examples, they are presented with a series of tests. As described in the instructions, each test displays two entries which were drawn from the TARGET, and one question which was drawn from the RETRO. The entries are drawn randomly without replacement throughout the survey, implying that the maximum number of tests a single annotator can be given,

$$N_{\text{test-max}} = \min\left(\frac{n_{\text{TARGET}} - 10}{2}, n_{\text{RETRO}} - 10\right), \tag{1}$$

where $n_{\text{TARGET}}$ and $n_{\text{RETRO}}$ are the number of entries in the TARGET and the RETRO, respectively.

If the RETRO is statistically indistinguishable from the TARGET, then human performance on this annotation test should not be statistically different from random selection.

For our results reported in REF, a total of 23 approved participants answered 230 trials to separate entries for the retro hold-out.

# F   Semantic Similarity

The code for this test was conducted entirely in Google Colab without any modifications to default settings, implying less than 12.7 GB of RAM and less than 107.7 GB of disk space used. Running the entire Jupyter Notebook in Google Colab takes approximately 1 hour to run using the free default runtime configuration.

Recall that PARENT is hypothetical parent distribution of entries from which TARGET and RETRO could be drawn independently. In an ideal scenario, we could determine the likelihood that RETRO was drawn from PARENT. Unfortunately, we do not have access to PARENT, so we need to get a bit creative. The largest dataset we have which could be representative of PARENT is (RETRO + TARGET). For this reason, we define a surrogate parent,

$$\text{PARENT}' := \text{RETRO} + \text{TARGET}.$$

We will then use PARENT′ in our tests to approximate the true PARENT. It is worth mentioning that, because of this approximation, TARGET and RETRO should have the same size. Unless the two datasets have the same number of entries, tests which leverage PARENT′ will require an initial random sub-sampling of the larger dataset, meaning that multiple iterations of this process will have to be leveraged.

To formally determine whether RETRO could belong to PARENT, we turn to the permutation test[3], a robust method for analyzing whether two distributions can be considered equivalent [14, 37]. For a true permutation test, we would use some test statistic to assess each unique subset of observations within PARENT′ that contains the same number of observations as RETRO. Formally, we define

$$\text{SUB} := \text{ a unique subset of PARENT}' \text{ with } n_G \text{ entries,}$$

where $n_G$ is the number of entries in RETRO. However, this quickly becomes infeasible for most meaningful test statistics due to computational complexity. More suited to our scenario is the random permutation test, in which the test statistic is calculated for $\text{SUB}_a \forall a \in [1, N]$ [22]. In the limit as $N$ approaches infinity, the result produced with a random permutation test will approach the result of a true permutation test.

Once we have our random samples, our next step is to calculate some test statistic for each of these samples, as well as RETRO; if our RETRO has an extreme score compared to the score of other SUBs, the test is indicating that it is less likely for RETRO to be drawn from PARENT′ than other possible samplings.

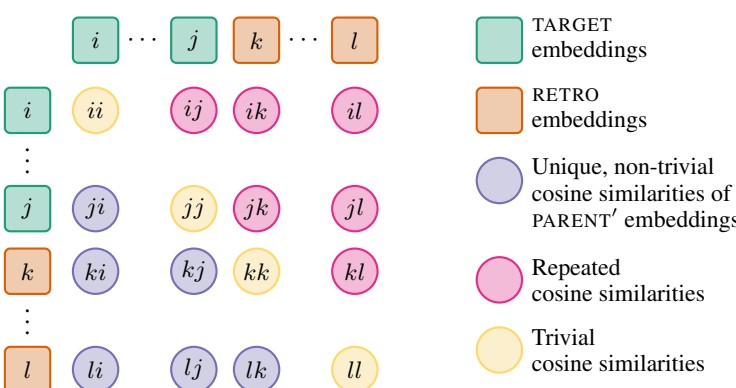

Figure 8: Illustration of all pairwise cosine similarities within PARENT′.

For our test statistic, cosine similarity between embeddings is a logical starting place because it is a tool that is frequently used in the field of Natural Language Processing as a baseline for sentence similarity [3], and it is a computationally efficient method for projecting the complex information stored in large embedding vectors down into a single variable. Details of embedding model usage are thoroughly documented in Appendix C.1.

---

[3]The Permutation Test: A Visual Explanation of Statistical Testing provides a good introduction to the test.

We can then convert the PARENT$'$ embeddings, which are multi-dimensional data, into analogous one-dimensional data by calculating all pairwise cosine similarities which are both unique and nontrivial.[4] The operation results in a normalized value which can be thought of as a measure for the similarity of meaning between two embedded sentences, with more similar phrases scoring close to one, and very different phrases scoring close to negative one.

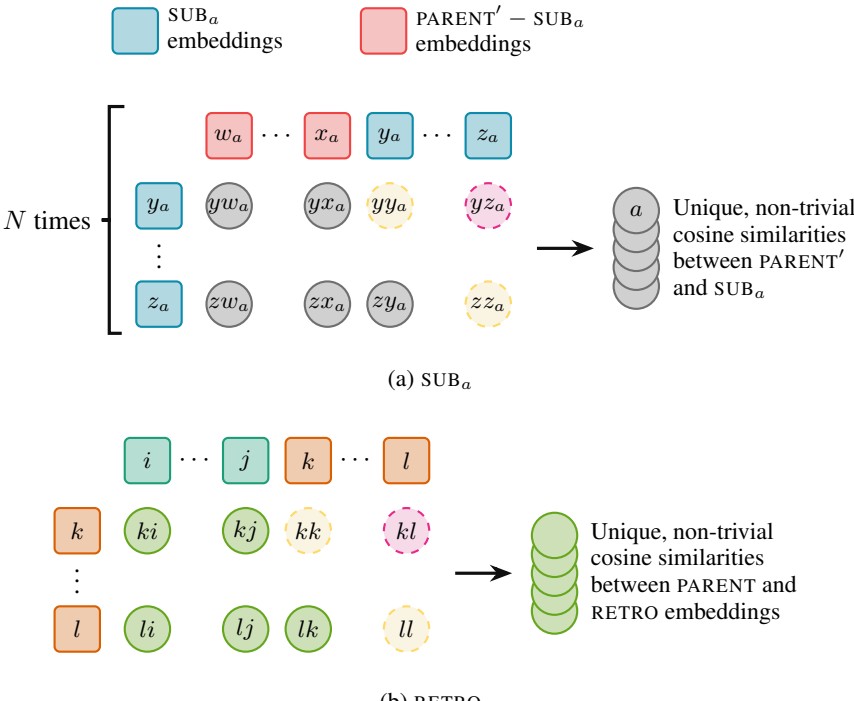

Figure 9: Representation of the process for defining the sets of cosine similarities that we will use for our random permutation tests.

Now that we have our sub-samples from PARENT$'$, which will serve as the population to which we compare RETRO, the random permutation test can begin.

The final step of this permutation test is to evaluate RETRO using the same test statistic and compare its value to that of our random samples. If the percentage of test statistics which are more extreme than the test statistic for RETRO is large ($p \geq 95\%$ or $p \leq 5\%$), the test indicates that our RETRO is an outlier along this particular metric, suggesting that it is less likely we can treat RETRO as a true holdout dataset for the TARGET.

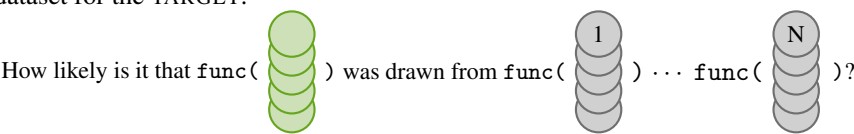

Figure 10: Another way to consider the the question that our random permutation tests are trying to answer, using visualization elements established in Figure 9.

---

[4]The cosine similarity between an embedding vector and itself is considered to be *trivial*.

# G Model Evaluation

## G.1 Model Experiments

Experiments were done through the OpenAI chat completion API as well by running various models from Huggingface with mostly default settings. Aside from generation length, we specified a temperature of 0.5, although it may be that OpenAI chat models do not use this parameter.

### G.1.1 Sampling

Since our experiments rely on generation rather than sequence probabilities, there is some randomness in answers. To address this and perform multiple samples until one answers stands out (minimum ten samples/questions, repeated until at least one option is ahead by three).

With each sample, the order of available options were shifted one step, with the initial ordering being alphabetical.

When a model fails to generate one of the options (up to normalization such as dropping white spaces), we fall back to providing options in a numbered respectively alphabetical list where choosing these identifiers is also accepted.

### G.1.2 Generation Prompt

For all models, a Vicuna-inspired prompt was used.

```
USER: This is a multiple-choice question. Answer it by repeating one of the
options, exactly and literally.
{question}
Available options:
{option_1}
{option_2}
..
{option_k}
Answer with one of the options.
ASSISTANT:
```

### G.1.3 Compute

Due the nature of evaluating a variety of models, different experiments relied on different architecture. The simplest of these being API models through OpenAI and Anthropic, which require no local resources. Other models were primarily hosted by Hugging Face. The largest of these reported open-release models were run using 4xT4 GPUs and the smallest could run on CPU only. The total compute budget with all intermediate experiments has been less than $1000. Evaluating a single model has cost between $1 and $50; and around 200 such experiments have been used to generate all the values and gaps used in this paper.

Additionally, the classifier-accuracy test does involve training a basic BERT model, although this is relatively quick on any consumer GPU.

