This document contains information about the supplementary material provided with submission number 2029 to the NeurIPS 2024 Benchmarks & Datasets track. Although the track intends for single-blind review, circumstances allowed us to submit completely anonymously, so we have opted for a double-blind submission.

# 1  Author Statement

The authors bear all responsibility in case of violation of rights.

*NOTE: The dataset does not yet have a DOI, as it has not yet been published. A DOI will be obtained once the dataset has been released.*

## 1.1  Licenses

- **Paper:** "Benchmark Inflation: Revealing LLM Performance Gaps Using Retro-Holdouts" is licensed under CC-BY-4.0.

- **Code:** The code used for experiments and data analysis is licensed under the MIT License.

- **Dataset:** Retro-TruthfulQA is licensed under CC-BY-SA-4.0.

# 2  Access

Due to our findings, we believe that all dataset creators should withhold a portion of their datasets to be continually re-evaluated to assess whether evaluation gaming is occurring. Accordingly, we have created two subsets of Retro-TruthfulQA: `public` and `private`. The `public` subset contains two thirds of the entries in the entire dataset; we plan to release it as v1.0.0 of the dataset, along with a pre-print of our work, in July of 2024. The `private` subset consists of the remaining third of the dataset – we plan to release v1.1.0, which will include both the `public` and `private` subsets, in May of 2025.

## 2.1  Access Instructions

To allow for reviewers to investigate the dataset and code, in full, prior to those releases, we created an anonymous HuggingFace account, which has uploaded a full version of the dataset and the codebase to two private repositories. These repos were then shared with an anonymous reviewer account, which all reviewers will share. We verified that no two-factor authentication is required. Below are the credentials for this HuggingFace account:

Username: `anonymous-reviewer`

Password: `2s6#4*h&57tJ%hrt`

Once you have logged into the account, you should be able to access the dataset and code at the following URLs:

Dataset: https://huggingface.co/datasets/peer-review/retro-truthfulqa_review

Code: https://huggingface.co/datasets/peer-review/benchmark-inflation-code_review/tree/main

These repositories can also be found by selecting the organization `Peer Review` on the left side of the screen.

# 3   The Dataset

## 3.1   Documentation

### 3.1.1   Croissant Metadata

Croissant Metadata were generated using the HuggingFace API endpoint. The file was then further edited to ensure appropriate documentation, and uploaded manually to the `retro-truthfulqa_review` repository as `croissant-v0.X.json`.

### 3.1.2   Datasheet

This PDF also contains the datasheet for Retro-TruthfulQA, as specified by Gebru et al. [1] in "Datasheets for Datasets". All names are redacted from this file, and repo links are not yet included, as the repos are not yet publicly available.

# 4   ML Reproducibility Checklist

This PDF also contains a filled out version of the ML reproducibility checklist, as described by Pineau et al. [2] in "Improving Reproducibility in Machine Learning Research (A Report from the NeurIPS 2019 Reproducibility Program)".

# References

[1] T. Gebru, J. Morgenstern, B. Vecchione, J. W. Vaughan, H. Wallach, H. D. I. au2, and K. Crawford. Datasheets for datasets, 2021.

[2] J. Pineau, P. Vincent-Lamarre, K. Sinha, V. Larivière, A. Beygelzimer, F. d'Alché Buc, E. Fox, and H. Larochelle. Improving reproducibility in machine learning research (a report from the neurips 2019 reproducibility program), 2020.

This document leverages the format introduced by Gebru et al. [1] in "Datasheets for Datasets" to properly document the Retro-TruthfulQA dataset. An example dataset datasheet can be seen in the appendix of the paper.

## Motivation

**For what purpose was the dataset created?** Was there a specific task in mind? Was there a specific gap that needed to be filled? Please provide a description.

> Retro-TruthfulQA was created as a retro-holdout dataset for the TruthfulQA benchmark in order to accurately assess the benchmark inflation that models exhibit on the public dataset.

**Who created the dataset (e.g., which team, research group) and on behalf of which entity (e.g., company, institution, organization)?**

> The dataset was created by the team behind the paper "Benchmark Inflation: Revealing LLM Performance Gaps Using Retro-Holdouts" – REDACTED.

**Who funded the creation of the dataset?** If there is an associated grant, please provide the name of the grantor and the grant name and number.

> The work was funded by REDACTED and the research team.

**Any other comments?**

> No.

## Composition

**What do the instances that comprise the dataset represent (e.g., documents, photos, people, countries)?** Are there multiple types of instances (e.g., movies, users, and ratings; people and interactions between them; nodes and edges)? Please provide a description.

Each instance represents a single multiple choice question designed to assess whether language models will parrot "human falsehoods" which are likely to have been found in training data [2].

**How many instances are there in total (of each type, if appropriate)?**

The entirety of Retro-TruthfulQA contains 817 entries.

As is discussed in the accompanying paper, analysis of TruthfulQA revealed there to be significant differences between the subsets marked as "Adversarial" and "Non-Adversarial" – accordingly, we use the same designation in our dataset by differentiating between which samples were based on "Adversarial" entries, and which were based on "Non-Adversarial" entries. The "Adversarial" category of the Retro-TruthfulQA dataset was not verified to be similar to its target, as the original OpenAI GPT-3 model which was used to adversarially filter the original TruthfulQA dataset has been discontinued.

| Release | Type | # |
|---------|------|---|
| `public` | Non-Adversarial | 253 |
| | Adversarial | 291 |
| `private` | Non-Adversarial | 127 |
| | Adversarial | 146 |

**Does the dataset contain all possible instances or is it a sample (not necessarily random) of instances from a larger set?** If the dataset is a sample, then what is the larger set? Is the sample representative of the larger set (e.g., geographic coverage)? If so, please describe how this representativeness was validated/verified. If it is not representative of the larger set, please describe why not (e.g., to cover a more diverse range of instances, because instances were withheld or unavailable).

The for-review version, v0.X, contains both the `public` split, consisting of two-thirds of the samples in the entire dataset, chosen at random from the larger set, as well as the `private` split. We plan to release the `public` split as v1.0.0 in July of 2024, and the entirety of the dataset as v1.1.0 in May of 2025. Prior to this release we will conduct further analysis on benchmark performance of both TruthfulQA and Retro-TruthfulQA.

In addition, dataset does not contain all possible instances of frequently repeated phrases that are likely to be used in language model training. The dataset was constructed to be a retro-holdout of the pre-existing TruthfulQA dataset, meaning that no TruthfulQA entry is verbatim in Retro-TruthfulQA.

We believe that our dataset is representative of the questions found in TruthfulQA, and this has be validated (see paper). We do not claim that the original TruthfulQA dataset, nor our Retro-TruthfulQA, are representative of all "human falsehoods".

1. The creation of TruthfulQA significantly leveraged online resources like Wikipedia to identify human falsehoods. This is likely not representative of all "human falsehoods" because it limits sources to those which were documented on the internet, and in English.

2. TruthfulQA includes questions which likely would not be classified as "human falsehoods" intuitively, such as entries in the category "indexical error" or "logical falsehood".

3. TruthfulQA has two "Types" of question: Adversarial and Non-Adversarial. An initial set was tested against GPT-3, and those which the model scored well on were removed, leaving the Adversarial entries. The Non-Adversarial entries were then made using the Adversarial as inspiration. Our analysis indicates that this process had a significant impact on the dataset; certain models with cut-off dates prior to the release of TruthfulQA score substantially worse (approximately $> 15\%$) on the Adversarial questions.

Retro-TruthfulQA intends to mimic the entries within TruthfulQA to such an extent that our new dataset can be treated as if it were part of the original distribution from which TruthfulQA questions originated, meaning that many of the failings of TruthfulQA will almost certainly be present in Retro-TruthfulQA.

**What data does each instance consist of?** "Raw" data (e.g., unprocessed text or images)or features? In either case, please provide a description.

Each entry consists of:

- **Release:** Designates the release which the entry will be published with.

- **Type:** The TruthfulQA "Type" of entry that was used as inspiration for the Retro-TruthfulQA entry.

- **Category:** The category of the question, e.g. Misconception, Indexical Error, Proverb.

- **Question:** A single question, written in English, and terminated with a question mark ("?").

- **Best Answer:** The correct multiple choice response as a string. This is the first entry in the **MC1 Targets** `dict`.

- **Incorrect Answers** Some number of incorrect multiple choice responses, as a list of strings.

- **MC1 Targets:** Dictionary including two to eleven possible multiple choice responses to the question, as well as correct/incorrect labels for each.

**Is there a label or target associated with each instance?** If so, please provide a description.

The target associated with each instance is the **Best Answer** field. Additionally, each response in MC1 Targets is labeled as either correct or incorrect within the entry itself. The first response listed is always the one which has been marked as correct by the creators of the Retro-TruthfulQA dataset, while all others are false. Each entry should contain exactly one MC1 Target marked as correct.

**Is any information missing from individual instances?** If so, please provide a description, explaining why this information is missing (e.g., because it was unavailable). This does not include intentionally removed information, but might include, e.g., redacted text.

No.

**Are relationships between individual instances made explicit (e.g., users' movie ratings, social network links)?** If so, please describe how these relationships are made explicit

No.

**Are there recommended data splits (e.g., training, development / validation, testing)?** If so, please provide a description of these splits, explaining the rationale behind them.

Because the Adversarial entries are not verified to be similar to those in TruthfulQA, we recommend making comparisons using only the Non-Adversarial subset.

Retro-TruthfulQA is designated as a testing split, as it should be used as a benchmark for models, and not used during training. This was chosen so that results on the benchmark between models can be somewhat comparable over time.

**Are there any errors, sources of noise, or redundancies in the dataset?** If so, please provide a description.

As there are a small number of spelling errors, inconsistencies, debatable entries, and redundancies, i.e. multiple questions probing virtually the same misconception, we aimed to capture these failure mode in our dataset as well. There are a small number of entries that could be considered as a paraphrasing of either another entry in Retro-TruthfulQA, or an entry in TruthfulQA. This decision was made in attempt to more accurately mimic the original dataset. There are no known unintentional errors in the dataset.

**Is the dataset self-contained, or does it link to or otherwise rely on external resources (e.g., websites, tweets, other datasets)?** If it links to or relies on external resources, a) are there guarantees that they will exist, and remain constant, over time; b) are there official archival versions of the complete dataset (i.e., including the external resources as they existed at the time the dataset was created); c) are there any restrictions (e.g., licenses, fees) associated with any of the external resources that might apply to a dataset consumer? Please provide descriptions of all external resources and any restrictions associated with them, as well as links or other access points, as appropriate

The dataset is self contained.

**Does the dataset contain data that might be considered confidential (e.g., data that is protected by legal privilege or by doctor–patient confidentiality, data that includes the content of individuals' nonpublic communications)?** If so, please provide a description.

No.

**Does the dataset contain data that, if viewed directly, might be offensive, insulting, threatening, or might otherwise cause anxiety?** If so, please describe why.

Although we took precautions to address this, there is a small chance that incorrect responses may be considered offensive/insulting to certain groups of people. It is important to note that these responses are labeled as incorrect, but we nonetheless understand that having these as possible responses at all could perpetuate stereotypes, misconceptions, etc. Our plan to address this is detailed in the Maintenance section of this datasheet.

**Does the dataset identify any subpopulations (e.g., by age, gender)?** If so, please describe how these subpopulations are identified and provide a description of their respective distributions within the dataset.

The data were not collected from populations, so the origin of the questions does not differ.

**Is it possible to identify individuals (i.e., one or more natural persons), either directly or indirectly (i.e., in combination with other data) from the dataset?** If so, please describe how.

Yes, there are a number of public figures which are mentioned by name throughout the dataset.

**Does the dataset contain data that might be considered sensitive in any way (e.g., data that reveals race or ethnic origins, sexual orientations, religious beliefs, political opinions or union memberships, or locations; financial or health data; biometric or genetic data; forms of government identification, such as social security numbers; criminal history)?** If so, please provide a description.

No.

**Any other comments?**

No.

## Collection Process

**How was the data associated with each instance acquired?** Was the data directly observable (e.g., raw text, movie ratings), reported by subjects (e.g., survey responses), or indirectly inferred/derived from other data (e.g., part-of-speech tags, model-based guesses for age or language)? If the data was reported by subjects or indirectly inferred/derived from other data, was the data validated/verified? If so, please describe how.

> Various online resources, such as Wikipedia, as well as three books [3–5] were leveraged for ideating possible questions, and entries within the original TruthfulQA dataset were used as a reference for word choice and phrasing within the new entries. Inspiration from personal experience of the dataset creators also informed a small portion of dataset entries. The process for generation of the dataset is described in detail in the accompanying paper.

**What mechanisms or procedures were used to collect the data (e.g., hardware apparatuses or sensors, manual human curation, software programs, software APIs)?** How were these mechanisms or procedures validated?

> The primary mechanism used for dataset creation was an interactive manual approach which leveraged iterative machine assisted analysis. In addition, entries from the Wikipedia page "List of common misconceptions" [6] were collected and sorted by date added; those which existed on the webpage prior to the release of TruthfulQA were not considered, as most were included in the original TruthfulQA dataset.

**If the dataset is a sample from a larger set, what was the sampling strategy (e.g., deterministic, probabilistic with specific sampling probabilities)?**

> The specific strategy was to randomize the order of each category, and then designate every third entry as `private`, while all others were designated as `public`.

**Who was involved in the data collection process (e.g., students, crowdworkers, contractors) and how were they compensated (e.g., how much were crowdworkers paid)?**

> Researchers on the project were the main contributors to data collection. In addition, REDACTED assisted in dataset curation for preliminary work, and were compensated REDACTED. REDACTED also contributed to the dataset with approximately 5 hours of time.

**Over what timeframe was the data collected?**

Data collection began during November of 2023, and continued through June 2024.

**Were any ethical review processes conducted (e.g., by an institutional review board)?** If so, please provide a description of these review processes, including the outcomes, as well as a link or other access point to any supporting documentation.

No.

**Did you collect the data from the individuals in question directly, or obtain it via third parties or other sources (e.g., websites)?**

No, any individual specifically named in the dataset is a public figure, and information were sourced from publicly available data.

**Were the individuals in question notified about the data collection?** If so, please describe (or show with screenshots or other information) how notice was provided, and provide a link or other access point to, or otherwise reproduce, the exact language of the notification itself.

No, information were sourced from publicly available data, and all individuals mentioned by name are public figures.

**Did the individuals in question consent to the collection and use of their data?** If so, please describe (or show with screenshots or other information) how consent was requested and provided, and provide a link or other access point to, or otherwise reproduce, the exact language to which the individuals consented.

N/A.

**If consent was obtained, were the consenting individuals provided with a mechanism to revoke their consent in the future or for certain uses?** If so, please provide a description, as well as a link or other access point to the mechanism (if appropriate).

If any individual were to request a specific question with their name in it be removed from the dataset using the process described in the maintenance section of this document, the entry would be removed from the dataset.

**Has an analysis of the potential impact of the dataset and its use on data subjects (e.g., a data protection impact analysis) been conducted?** If so, please provide a description of this analysis, including the outcomes, as well as a link or other access point to any supporting documentation.

N/A.

**Any other comments?**

    No.

## Preprocessing/cleaning/labeling

**Was any preprocessing/cleaning/labeling of the data done (e.g., discretization or bucketing, tokenization, part-of-speech tagging, SIFT feature extraction, removal of instances, processing of missing values)?** If so, please provide a description. If not, you may skip the remaining questions in this section.

    Yes, this process is fully documented in the accompanying paper, "Benchmark Inflation: Revealing LLM Performance Gaps Using Retro-Holdouts" – it was conducted in order to maximize Retro-TruthfulQA's similarity to that of TruthfulQA.

**Was the "raw" data saved in addition to the preprocessed/cleaned/labeled data (e.g., to support unanticipated future uses)?** If so, please provide a link or other access point to the "raw" data.

    No.

**Is the software that was used to preprocess/clean/label the data available?**

    Yes.

**Any other comments?**

    No.

## Uses

**Has the dataset been used for any tasks already?** If so, please provide a description.

    Yes, it was used to quantify benchmark inflation of models on the TruthfulQA benchmark. The results are fully described in the accompanying paper.

**Is there a repository that links to any or all papers or systems that use the dataset?** If so, please provide a link or other access point.

    Yes, please refer to documentation at `see-pdf-access-instructions`.

**What (other) tasks could the dataset be used for?**

The dataset could be used as an extension of, or stand-in for, the original TruthfulQA dataset, which is intended to assess the extent to which language models mimic "human falsehoods". The dataset can also be used as an example of what a retro-holdout dataset might look like.

**Is there anything about the composition of the dataset or the way it was collected and preprocessed/cleaned/labeled that might impact future uses?** For example, is there anything that a dataset consumer might need to know to avoid uses that could result in unfair treatment of individuals or groups (e.g., stereotyping, quality of service issues) or other risks or harms (e.g., legal risks, financial harms)? If so, please provide a description. Is there anything a dataset consumer could do to mitigate these risks or harms?

Because the data were selected/ideated by English speakers in North America and Europe, it is likely that the distribution of misconceptions skews towards the ones held in those regions.

**Are there tasks for which the dataset should not be used?** If so, please provide a description

The dataset **should not be used at any point during the training process**. Let's see how long that lasts.

**Any other comments?**

No.

## Distribution

**Will the dataset be distributed to third parties outside of the entity (e.g., company, institution, organization) on behalf of which the dataset was created?** If so, please provide a description.

Yes, the dataset will be publicly available on the internet.

**How will the dataset will be distributed (e.g., tarball on website, API, GitHub)?** Does the dataset have a digital object identifier (DOI)?

The dataset is hosted on HuggingFace at `see-pdf-access-instructions`, and on OpenML at `see-pdf-access-instructions`. Once published, the dataset will have a DOI, but it cannot have a DOI until it has first been published. This document will be updated at that time.

**When will the dataset be distributed?**

The public split of the dataset, v1.0.0, will first be distributed with the pre-print of the accompanying paper in July of 2024. The private split will be added to the hosted dataset in May of 2025.

**Will the dataset be distributed under a copyright or other intellectual property (IP) license, and/or under applicable terms of use (ToU)?** If so, please describe this license and/or ToU, and provide a link or other access point to, or otherwise reproduce, any relevant licensing terms or ToU, as well as any fees associated with these restrictions.

The dataset is licensed under CC-BY-SA 4.0.

**Have any third parties imposed IP-based or other restrictions on the data associated with the instances?** If so, please describe these restrictions, and provide a link or other access point to, or otherwise reproduce, any relevant licensing terms, as well as any fees associated with these restrictions

No.

**Do any export controls or other regulatory restrictions apply to the dataset or to individual instances?** If so, please describe these restrictions, and provide a link or other access point to, or otherwise reproduce, any supporting documentation.

No.

**Any other comments?**

No.

## Maintenance

**Who will be supporting/hosting/maintaining the dataset?**

The corresponding authors of the accompanying paper, REDACTED will be supporting and maintaining the dataset. The dataset will be hosted on HuggingFace and OpenML.

**How can the owner/curator/manager of the dataset be contacted (e.g., email address)?**

Email: REDACTED

**Is there an erratum?** If so, please provide a link or other access point.

Currently no erratum is provided, but we plan to include an erratum on both dataset repositories beginning with v1.0.0.

**Will the dataset be updated (e.g., to correct labeling errors, add new instances, delete instances)?** If so, please describe how often, by whom, and how updates will be communicated to dataset consumers (e.g., mailing list, GitHub)?

Yes, if any of the following are brought to our attention, the issue will be rectified and the dataset re-released:

- If we are made aware of labeling errors, they will be corrected.

- If any persons explicitly mentioned within the dataset submit a request to have entries referring to them removed, the entries will be removed.

For all other requests regarding dataset entries, decisions will be made by the dataset maintainers on a case-by-case basis.

In addition, the private split of the dataset will be released as v1.1.0 in May of 2025. We do not intend to re-release the dataset for any other reason at this time.

Dataset consumers will be notified via HuggingFace in the event of a re-release.

**If the dataset relates to people, are there applicable limits on the retention of the data associated with the instances (e.g., were the individuals in question told that their data would be retained for a fixed period of time and then deleted)?** If so, please describe these limits and explain how they will be enforced.

No, all information relating to specific individuals refer to public figures and publicly available information.

**Will older versions of the dataset continue to be supported/hosted/maintained?** If so, please describe how. If not, please describe how its obsolescence will be communicated to dataset consumers.

Once the private split of Retro-TruthfulQA is released, both the public and private splits will be maintained. In the event that the dataset is updated either to correct a labeling error or remove entries at the request of any persons, the prior datset will not be maintained.

**If others want to extend/augment/build on/contribute to the dataset, is there a mechanism for them to do so? I** If so, please provide a description. Will these contributions be validated/verified? If so, please describe how. If not, why not? Is there a process for communicating/distributing these contributions to dataset consumers? If so, please provide a description.

We do not plan to accept additional entries to the dataset at this time, although passionate teams should contact us with their ideas either via email or HuggingFace.

**Any other comments?**

No.

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

  See first page for login credentials

For an **theoretical claim**, check if you include:

☑ A statement of the result.

☑ A clear explanation of any assumptions.

☑ A complete proof of the claim.
  Yes. All claims that require proofs have simple but complete proofs - e.g. whether a p-value is above or below a threshold.

For all **figures** and **tables** that present empirical results, check if you include:

☑ A complete description of the data collection process, including sample size.
  "Non-Adversarial" dataset size:    public = 253    private = 127
  "Adversarial" dataset size:        public = 291    private = 146

☑ A link to a downloadable version of the dataset or simulation environment.
  `https://huggingface.co/datasets/peer-review/retro-truthfulqa_review`
  See first page for login credentials

☑ An explanation of how samples were allocated for training / validation / testing.

☑ The range of hyper-parameters considered, method to select the best hyper-parameter configuration, and specification of all hyper-parameters used to generate results.
  - No hyperparameter optimization has been conducted.
  - §2.3 (Difficulty Similarity Test)
    Hyperparameters for classification accuracy used the huggingface defaults. Amplifications methods were used with top-k in the range 1-5 and fewshot in the range 0-5.

✅ The exact number of evaluation runs.
Throughly documented in the accompanying paper. For convenience, a summary of runs using our evaluation harness is documented here:

- For each evaluated model (Fig. 3b) the model was assessed once on TruthfulQA, and once on Retro-TruthfulQA using our custom evaluation harness. Within the harness itself, each question was independently queried between 10 and 200 times.

- In addition, four models were assessed five times on both benchmarks to quantify error of our evaluation harness. These models were `davinci-002`, `babbage-002`, `NeoX-20B`, and `GPT-3.5`.

✅ A description of how experiments were run.

✅ A clear definition of the specific measure or statistics used to report results.

✅ Clearly defined error bars.

✅ A description of results with  central tendency (e.g. mean) and  variation (e.g. stddev).

✅ A description of the computing infrastructure used.

# References

[1] J. Pineau, P. Vincent-Lamarre, K. Sinha, V. Larivière, A. Beygelzimer, F. d'Alché Buc, E. Fox, and H. Larochelle. Improving reproducibility in machine learning research (a report from the neurips 2019 reproducibility program), 2020.