# OpenReview forum: "Benchmark Inflation: Revealing LLM Performance Gaps Using Retro-Holdouts"
_NeurIPS.cc/2024/Datasets_and_Benchmarks_Track — Submitted to NeurIPS 2024 Track Datasets and Benchmarks_

### Official Review · Reviewer_dLqB · 2024-06-20
**Useful Contribution and Methodology, Improvable with Further Discussions**

**Rating:** 6
**Confidence:** 4

**Review:**

Pros:
- The procedure introduced for creating retro-holdouts will surely be useful to the community and inspire further work in this direction.
- The procedure is comparatively thorough.
- The benchmark dataset when released will surely be useful to the community.


Cons:
- Overall, the work would benefit from further exploration and analysis of model performance and perceived evidence of evaluation gaming. While evaluation results on a wide range of models are provided, there is not much discussion of which models have been found to overfit
- Some charts are unclear. For example, in Figure 4, the meaning of "Benchmark Inflation" is not particularly clear--is a positive, or a negative value preferable? This can be determined through cross referencing the previous page's accuracy plot, but is not intuitive.


I think that this paper will be a useful contribution to the community.

**Strengths:**

This paper puts forward a methodology that can be followed by other researchers to create holdout datasets, that can be used to test for evidence of evaluation gaming. It will be of interest to a large portion of the language modeling community, including those who study evaluation, those who develop and fine-tune new models, and those who build new language models from scratch. It has positive ethical and social implications by having the potential to expose overclaimed results of new language models and serve as a warning signal for overfitting.

**Additional Feedback:**

Appendix A should be cleaned up for presentation at the conference--it appears to be internally-facing dataset construction instructions. Providing a copy of such instructions is a useful supplement, but should be more clearly marked within the appendix text.

**Clarity:**

The paper is mostly well-written. Some sections, such as the appendix, would benefit from extra proofreading and improved formatting/clarity.

Some figures could be presented better, as mentioned above.

**Correctness:**

The procedure for producing a new retro-holdout dataset appears sound and effective.


However, the authors' evaluation procedure (G.1.2) for the TruthfulQA task appears nonstandard. For example, the Vicuna prompt is insufficiently motivated--more discussion of how this might skew results would be beneficial.

**Documentation:**

Yes, there is enough information documenting the dataset and its creation process.

**Ethics:**

No, I do not anticipate serious ethical concerns.

**Limitations:**

Yes, the authors have addressed potential limitations of their work. The work does not have clear negative social impacts or risks.

**Opportunities For Improvement:**

As mentioned above, the paper could be improved both by adding further experiments analyzing performance on the dataset, and by more thoroughly discussing the results of usage of the retro-holdout dataset. The presentation of the paper, such as in some of its visuals' clarity, could also be improved.

**Relation To Prior Work:**

This work does discuss prior work. The most similar is the concurrent work by Zhang et al. [1], but the authors' proposed procedure for creating such retro-holdouts is more rigorous than that followed in [1].



[1] Zhang et al., 2024. "A Careful Examination of Large Language Model Performance on Grade School Arithmetic".  https://arxiv.org/abs/2405.00332

**Summary And Contributions:**

This work presents two major contributions: 1) a procedure for creating statistically-indistinguishable retro-holdout evaluation test sets and 2) a result of following that procedure, Retro-TruthfulQA. Most of the Retro-TruthfulQA dataset will be made available publicly, with an additional holdout portion committed to being released in 1 year.

---

> ### Author Rebuttal · Authors · 2024-08-19
>
> Reviewer dLqB, we thank you for your review, and address your points below:
>
>
> > Overall, the work would benefit from further exploration and analysis of model performance and perceived evidence of evaluation gaming. While evaluation results on a wide range of models are provided, there is not much discussion of which models have been found to overfit
>
> We agree that more analysis and takeaways of our work would be beneficial, and plan on addressing this in the upcoming version of our work. Some relevant items to mention are the levels of contamination seen in the CONDA data contamination database for the TruthfulQA dataset, and the correlation between OpenAI's models being able to replicate outputs from TruthfulQA and the model's benchmark inflation scores.
>
> We will also make it more clear which models we can be certain have gamed the TruthfulQA evaluation. This is partly addressed in a revision of our Figure 4, which we believe is a substantial improvement.
>
> > Some charts are unclear. For example, in Figure 4, the meaning of "Benchmark Inflation" is not particularly clear--is a positive, or a negative value preferable? This can be determined through cross referencing the previous page's accuracy plot, but is not intuitive.
>
> We plan on making significant changes to our plots to increase their clarity. A first version of this can be seen in the attached PDF, which includes a newer version of our Figure 4. In addition to the changes seen here, we also plan to add annotations directly to the chart to make it clear how the inflation value is being calculated, and what the error bars signify.
>
> > However, the authors' evaluation procedure (G.1.2) for the TruthfulQA task appears nonstandard. For example, the Vicuna prompt is insufficiently motivated--more discussion of how this might skew results would be beneficial.
>
> Our primary goal is to establish that there are statistically significant gaps between public benchmark scores and what would be expected from hold-out data. As such, there is no one correct prompt and in fact, if there are gaps with any prompt, that demonstrates benchmark inflation.
>
> The failure mode with the prompt is rather that some models may fail to even perform the task and so will perform at the level of random answers for both the public and private datasets.
>
> We have experimented with a few prompt variants to avoid this behavior and actually get non-trivial performance also on older models. We have not noticed the effect that models have benchmark inflation with some prompts but not others, other than when the prompts cause the models to perform no better than random.
>
> Crucially, in our case, we need a prompt that works with both more modern conversational models and older continuation-based models. The Vicuna prompt is one of the most popular and has shown itself suitable for this. We have adapted it for multiple-choice question, as that is the task that is being tested. We are also not aware of any prompt that could be considered more standard.
>
> We believe you are correct however that some kind of sensitivity analysis due to the prompt would be instructive and will include one, but focusing only on a few key models.
>
> > The paper is mostly well-written. Some sections, such as the appendix, would benefit from extra proofreading and improved formatting/clarity.
>
> We appreciate this callout, and will ensure that a thorough review of the appendices, both current and not yet added, will be completed.
>
> --
>
> Lastly, we want to clarify that we see our primary contribution is the demonstration that evaluation gaming, such as the data leakage which has been well documented in other works, is having a quantifiably significant impact on reported benchmark scores

---

> > ### Comment · Reviewer_dLqB · 2024-08-26
> > **Response**
> >
> > I thank the authors for their response. The new Figure 4 is indeed a substantial improvement, and analyzing the relevance of reported contamination levels for TruthfulQA to cross-reference against the experimental Retro-TruthfulQA results would be a strong addition if pursued.
> >
> >
> > I will maintain my score.

---

> > ### Author Response · Authors · 2024-08-31
> > **Thanks**
> >
> > Thank you Reviewer, for your detailed feedback and for taking the time to go through our revisions. We were also excited to hear that you would have wanted further exploration of the implications on models and we will try to include some aspect of this.
> >
> > We really value your insights and the constructive points you’ve raised. Thanks again for helping us strengthen the paper.

---

### Official Review · Reviewer_sHLe · 2024-07-25
**Interesting idea and dataset; descriptions overly vague in places**

**Rating:** 6
**Confidence:** 3
**Clarity:** Yes, the paper is well written with m…

**Review:**

This paper presents an interesting prospective solution to the problem of overfitting to evaluation datasets in ML. The work is original, of good quality, and of potential significance to the ML community.

The primary strength of this paper is that it makes a convincing argument that some kind of overfitting/evaluation gaming is plaguing the TruthfulQA benchmark. The authors went through great care and substantial effort to create the retrospective dataset that they use to demonstrate this point. This is an important contribution to better understanding evaluation shortcomings in ML and should inspire future research and initiatives to address this issue.

However, the paper falls short of providing a useful roadmap for other members of the community who would like to create their own retrospective datasets for benchmarks outside of TruthfulQA, or outside of text datasets. In particular, exact specifications of the dataset construction process, which is seemingly largely manual, are described quite vaguely in the text (more details provided later in the review).

Furthermore, I would like to see more discussion from the authors on why it is worthwhile to go through their process of constructing a retrospective dataset rather than trying to emulate the original dataset construction process itself, i.e., collecting more data. Collecting real data should be the gold standard, but obviously, it can be a quite expensive process. It should be made clearer in the paper why the authors’ proposed process is advantageous, or if it is not always advantageous, in which situations it makes sense for one to follow the authors’ process or to try to collect more original data from the source.

**Strengths:**

- The paper makes a very convincing case for the presence of benchmark inflation in TruthfulQA.
- The tests used to test that a retrospective dataset is suitable are thoughtfully constructed and could be useful for followup work.

**Additional Feedback:**

If more detailed descriptions of the entry formulation and iterative processes are added, I am willing to raise my score above the acceptance threshold.

**Correctness:**

Excluding any points raised above, to my knowledge, the claims made in this paper are correct.

**Documentation:**

A URL for reviewer access, hosting plan, license, and maintenance plan are all provided in the Supplementary Material. Structured documentation is provided via Croissant documentation and a datasheet. As discussed above, the specifics of the entry forumation and iterative process are described in a vague manner such that it is unclear whether there is sufficient detail to support reproducibility.

**Ethics:**

No, the paper does not warrant further ethical review.

**Limitations:**

The authors discuss the limitations of their work in Section 3.5. Excluding the points discussed above, the authors have sufficiently addressed these limitations. Other than inadvertently perpetuating the biases of the original dataset (which was mentioned by the authors in the paper), I do not see other negative societal impacts of this work.

**Opportunities For Improvement:**

- Section 2.1, Build Intuition: This section appears to basically just advise that one familiarize themselves with the target dataset, but doesn’t provide any actionable information on what one needs to know in order to create a retrospective dataset, other than reading documentation and looking through many examples. Concretely, what should one be doing in this phase of the process?

- Section 2.1, Entry Formulation: This part of the process remains quite opaque given only the text provided in the main paper. It is not even clear immediately from the text that this is a manual process in which humans are generating the new instances. Despite the limited page constraint, this seems like a major point to leave largely undiscussed in the main paper. While there is an extensive description of the process and tools used in Appendix A, it is not discussed how one can adapt this procedure for other datasets other than TruthfulQA. Nor does it appear to be discussed how many people were involved in generating examples for Retro-TruthfulQA.

- Section 2.2: While these tools might provide interesting aggregate insight into how well the process is working, it remains ambiguous how these tools inform the iterative process, i.e., if I see that my retro dataset is off as determined by visual inspection using these tools, it seems like it would challenging to interpret what I need to do with my dataset to remedy this issues. The iterative process is largely undiscussed in both the main paper and the appendix, but is quite a large part of this procedure. This is a major barrier to those who would like to adapt the procedure to their own use cases.

- Section 2.3, Similarity of difficulty: The specifics of this test do not appear to actually be specified anywhere, including in the main text and in Appendix D. Is this a hypothesis test? When is the difficulty “close enough”?

- Section 2.3, Prediction accuracy: Similarly here, when is the accuracy “close enough” to 50%? What statistical test is being performed here?

- Section 2.3, Semantic Embedding Similarity + Appendix F: It seems as though the authors are essentially constructing a mixture distribution as a proxy for the parent distribution and then testing whether retro could belong to this mixture. Intuitively, this seems like it will roughly work, but I wonder if there is precedent for this kind of comparison, and if the p-value is really meaningful under these conditions. Is there anything that the authors could point to in the literature to provide further justification and understanding as to the statistical validity of this procedure?

- Section 2.3.1: Even given this section, the descriptions of how the iterative process is conducted concretely are left opaque, e.g., how many iterations did the process take?

- Section 3.1, lines 236-237: Why is Retro-TruthfulQA only applicable to models with a training cutoff before Jan 1, 2024? I understood this point in evaluating the similarity of difficulty, but now that the dataset has been constructed, shouldn’t it be applicable to any model that has not seen it? I may have missed something here.

- Figure 4: Where does the uncertainty come from in this plot?

- Section 3.4: Although this is interesting, it feels a bit lengthy and out of place in the main paper, as it is not part of the main contributions of this paper. Perhaps it could be moved to the appendix and discussed more at length there?

- Section 3.5: Also, are these techniques only applicable to text datasets? If not, how easy would it be to adapt this process to non-text datasets?

- Section 3.5: Since this process is resource-intensive, is the process of creating a retro-holdout dataset better than trying to collect new, original data from the source? Why or why not?

**Relation To Prior Work:**

Yes, to my knowledge, it is sufficiently discussed how this work differs from previous contributions.

**Summary And Contributions:**

In this paper, the authors retroactively create a holdout dataset intended to come from the same parent distribution as TruthfulQA; they name this dataset Retro-TruthfulQA. The dataset is constructed using a manual, iterative process that is aided and tested by various tools introduced by the authors. They evaluate 20 LLMs on Retro-TruthfulQA and demonstrate that many LLMs have lower scores on this benchmark than on the original test dataset, demonstrating that some kind of overfitting, or evaluation gaming, is occurring.

---

> ### Author Rebuttal · Authors · 2024-08-19
>
> Thank you for the constructive feedback. We address each key point below:
>
> **Section 2.1, Build Intuition:**
>
> We understand the concern about limited actionable information. Our paper focuses on demonstrating the feasibility of retro-holdout datasets and quantifying the gap between public test scores and real performance. We will improve clarity by revising Figure 1, Section 2, and adding more details in the appendix including a high-level process. To summarize: understand original process, identify common patterns, brainstorm ideas, pair original entries and use same syntax, run tests and iteratively address differences.
>
> **Section 2.1, Entry Formulation:**
>
> We appreciate the specific feedback. We will clarify the manual nature of the dataset construction. All steps, except for analysis tools, were conducted manually to avoid LLM dependencies as possible gap explanations. We will also add an appendix that offers a more concrete 'roadmap' for replicating this process with other datasets.
>
> **Section 2.2, Tools and Iterative Process:**
>
> We acknowledge the ambiguity in how the tools inform the iterative process. We will clarify how visual inspection and tools guided our iterative refine While we do not believe that creating retro-holdout datasets is purely mechanistic, we will include practical advice on how to approach this process for different datasets and provide insights into the challenges we encountered. We note that our main goal was to demonstrate feasibility and establish score gaps
>
> **Section 2.3, Similarity of Difficulty:**
>
> We recognize the need for more detailed test explanations and have included them. Fisher’s Exact Test was used with 95% CI. If all pre-release models fall within the interval, we conclude that the difficulties are not sufficiently different. This formalization is added to an appendix, along with a clearer explanations for each test.
>
> Most importantly, an appendix now formalizes what it means for the dataset to be indistinguishable, derived from the random split of hold-out datasets, and show that the tests we have are different attempts at rejecting that hypothesis.
>
> **Section 2.3, Prediction Accuracy:**
>
> This is a test that has been used e.g. in Fake It Till You Make It: Guidelines for Effective Synthetic Data Generation.If the sets are indistinguishable, a classifier should not differentiate between them. Similarly the human annotation test is a triangle test. This produces binomial tests.
>
> **Section 2.3, Semantic Embedding Similarity + Appendix F:**
>
> We apologize for the lack of clarity here. As mentioned, the formalizing appendix should address this. We believe that the test directly attempts to reject the hypothesis and that the p value indeed is meaningful. We have removed any references to the parent distribution to clarify. The indistinguishability property derives directly from testing if a dataset could have been split randomly. There is no one correct test for this hypothesis and we have chosen a few.
>
> We note that while the components have been used before, the explicit claim that the dataset is indistinguishable is novel and the tests are used to attempt to reject this hypothesis. We believe that this is part of our novel contributions. Tests like the prediction accuracy have been used in synthetic data generation. The comparison of datasets using cosine similarities is also common but often omit the rigor of a statistical test. An example that has attempted it is "Synthetic Data Generation with Large Language Models for Text Classification: Potential and Limitations".
>
> **Section 2.3.1, Iterative Process Details:**
>
> We will include more information about the iterative process, such as the number of iterations required for different categories within TruthfulQA. This detail, while varying significantly across categories, will be summarized to provide a clearer picture of the process without overwhelming the reader with unnecessary minutiae.
>
> **Section 3.1, Applicability Date of Retro-TruthfulQA:**
>
> This is simply a recognition of when parts of our dataset became available online.
>
> **Figure 4:**
>
> The source of uncertainty in this figure, which comes from the sample size differences between the public and retro holdout datasets, will be clarified. The revised figure will include single sigma error bars and p-values, ensuring that the reader understands the statistical significance of the results. See attached.
>
> **Section 3.4, Length and Placement:**
>
> We agree that this section may be too lengthy for the main paper. We have moved this discussion to the appendix, allowing us to focus more on the primary contributions in the main text while providing a more detailed discussion for interested readers.
>
> **Section 3.5, Applicability to Non-Text Datasets and Comparison to Extending Datasets:
> **
>
> We will expand the discussion on this. In short, should be for all labeled datasets but other tooling. Extending is often the right answer for the field. However, to prove the score gap being due to public-data contamination, we need to mimic the original rigorously. Extending instead improves from a dataset intention POV.
>
> **Roadmap and Additional Details:**
>
> We will include an improved construction roadmap in the appendix, outlining steps such as understanding the original dataset, gathering new ideas, pairing them with original samples, and conducting iterative refinements using the provided tools. This roadmap will guide researchers in creating retro-holdout datasets for various benchmarks.
>
> --
>
> We see our most significant contribution as demonstrating that retro evaluation gaming is a significant and quantifiable issue in public benchmarks. Retro-holdouts offer a practical solution to this and while we have focused on TruthfulQA, the principles should extend to other domains.
>
> We appreciate your constructive feedback and believe these revisions will significantly strengthen our paper. Thank you for your time and consideration.

---

> > ### Author Response · Authors · 2024-08-27
> >
> > Reviewer sHLe,
> >
> > As we near the end of the rebuttal period, we wanted to verify our responses address all of your concerns with our submission. If there are remaining doubts beyond what’s discussed in our rebuttals, we want to ensure that we still have time to resolve them.
> >
> > We believe that the formalism of our statistical tests is a critical aspect of your review for us to address, so we will also include an excerpt from the rewritten section of our paper detailing the semantic similarity permutation test, as well as the appendix which goes into further detail.
> >
> > We know that things are likely quite busy for you currently, so any thoughts, even just preliminary ones would be greatly appreciated in helping us prioritize potential remaining improvements.
> >
> > #### Short description, main paper
> >
> > We perform a permutation test using semantic embeddings from Sentence Transformers \citep{fisher_design_1974, normaldeviate_modern_2012, hemerik_term_2024}. We let the test statistic be the mean of all cosine similarities between embeddings of dataset entries. The statistic of TARGET and RETRO is compared to random permutations of the same elements. As discussed in \Cref{app:formalization}, since the number of permutations is great, this approximated through Monte Carlo sampling of permutations and with sufficiently many samples, provides a tight bound. Details for test are described in \Cref{app:semantic-embedding}.
> >
> > #### Fully formalized test, appendix
> > Included in following comment. Apologies for non-standard formatting in this, we were not able to leverage the math fonts (rm, bb, bold, etc.) within this input, so we resorted to using accessible characters.

---

> > > ### Author Response · Authors · 2024-08-27
> > >
> > > (Had some more difficulties with LaTeX formatting here, so we just removed it for this version)
> > >
> > > #### Hold-out testing formalization
> > >
> > > In this appendix, we define what it means that a retroactively constructed dataset could have
> > > been a holdout set for a public dataset, as well as how this can be formalized and statistically tested.
> > >
> > > We define a labeled dataset D as a set of tuples (x, y) ∈ X × Y [..]
> > >
> > > We rely on the standard assumption in machine learning that a constructed dataset consists of i.i.d.
> > > samples from some distribution [25, 54].
> > >
> > > Given a dataset D, we define a public-holdout split of sizes (n_p, |D| − n_p) as the random variables
> > > D_p and D_h [..]
> > >
> > > In contrast to the regular setting, we can not assume that all of D has been drawn independently from
> > > the same distribution. Instead, D_p and D_h were constructed separately. [..] The claim that we want to
> > > show is that for our retro holdout dataset, D_p = D_h.
> > >
> > > We will design a number of statistical tests to attempt to reject this hypothesis. We
> > > will both employ various binomial tests for this, as well as a permutation test.
> > > Notably, the expected accuracy on both sets are statistically close, provided that a function f is
> > > independent of these samples. [..]
> > >
> > > Hence, given that one can show that the retro holdout dataset could have been drawn from the same
> > > distribution as the public dataset, and the difference in the accuracies is greater than some bound,
> > > then the remaining difference must be due to direct or indirect exposure to the public data.
> > >
> > > #### Permutation tests
> > >
> > > Given two sets A_i ⊆ (X × Y)^|A_i| for i = 1, 2, and a test statistic g : (X × Y)^(|A1|+|A2) → R
> > > which is invariant under permutation of the first |A_1| elements as well as the last |A_2| elements,
> > > and where A_i ∼ D_i^|A_i| for some distributions D_i, a permutation test is a test for the
> > > null hypothesis that D1 = D2.
> > >
> > > The p-value of a permutation test is the probability that the test statistic g is at least as extreme
> > > as the observed value under the null hypothesis. That is, let π_1, . . . , π_m be all permutations of
> > > A1 ⊎ A2 and for a permutation π, let A_{1,π}, A_{2,π} be the first |A_1| and last |A_2| elements of π.
> > > Let the average statistic be µ = E_π [g(A_{1,π}, A_{2,π})]. Then the two-sided p-value of the hypothesis
> > > given the observed statistic g(A1, A2) is given by P_π(|g(A_1, A_2) − µ| ≤ |g(A_{1,π}, A_{2,π}) − µ|).
> > > Since the number of permutations can be large, one can use a Monte Carlo approximation to estimate
> > > the p-value through sampling [12]. If N independent samples produce a p-value estimate of q, then a
> > > 99% confidence interval for the p-value is given by q ± 2.807 · q(1 − q)/√N.

---

> > > > ### Comment · Reviewer_sHLe · 2024-08-29
> > > > **Response to author rebuttal**
> > > >
> > > > Thank you for your detailed reply to all of my comments. I have raised my score above the acceptance threshold in light of the author's replies.

---

> > > > > ### Author Response · Authors · 2024-08-31
> > > > > **Thank you for the very detailed feedback!**
> > > > >
> > > > > Thank you revewer for taking the time to thoroughly engage with our work and for your extensive and highly useful feedback. Especially both the point on understanding both the formal approach and how it can be replicated. We’re glad to hear that our responses helped clarify things and that you found our revisions satisfactory.
> > > > >
> > > > > We really appreciate your constructive comments—they’ve definitely contributed to improving the paper. Thanks again for your time and for raising your score.

---

### Official Review · Reviewer_1HP3 · 2024-07-25
**Review of Benchmark Inflation**

**Rating:** 4
**Confidence:** 4
**Correctness:** Yes
**Clarity:** The paper requires proof read, especi…

**Review:**

Fair LLM evaluation is critical in the field. The authors have designed a process to tackle the problem based on the assumption that a hold-out dataset shares the same distribution as the original dataset. However, this does not apply to most benchmarks, especially for those with multiple categories with different data sources. In the paper, the authors only validate their method's effectiveness on TruthfulQA with certain evaluation types and categories.

**Strengths:**

1. A systematically designed way to construct a retro-holdout benchmark
2. Leverage existing benchmarks to reevaluate the LLM for fair comparison

**Additional Feedback:**

As listed in the Opportunities For Improvement section.

**Documentation:**

The authors have provided the link to the Hugging Face dataset.

**Ethics:**

No.

**Limitations:**

1. The authors assume that a generated hold-out dataset still possesses the same distribution as the original dataset. If the assumption does not stand, the evaluation result from the generated dataset may lose its original purpose, which is to make a fair evaluation.
2. The authors only validate their proposed method on one benchmark.

**Opportunities For Improvement:**

I suggest the authors provide more evidence on other datasets and prove the proposed method can be generalizable to other benchmarks.
In addition, the paper requires proof reading and is not easy to follow.

**Relation To Prior Work:**

Yes

**Summary And Contributions:**

The paper presents a method to systematically build a holdout dataset for fair LLM evaluation. The authors made the following contributions:
1. A systematic way to construct retro-holdout benchmark based on existing one
2. An experiment to validate the effectiveness of the proposed method by comparing TruthfulQA and the constructed Retro-TruthfulQA

---

> ### Author Rebuttal · Authors · 2024-08-19
>
> Reviewer 1HP3, we thank you for your review, and address your points below:
>
> > "...based on the assumption that a hold-out dataset shares the same distribution as the original dataset. However, this does not apply to most benchmarks, especially for those with multiple categories with different data sources." > "The authors assume that a generated hold-out dataset still possesses the same distribution as the original dataset."
>
> You are correct that this may be the case for real hold-out datasets. Especially ones design to e.g. assess out-of-distribution generalization.
>
> We are however not using existing real holdout datasets. Rather, we simply rely on the more narrow definition of holdout sets, in which the holdout is a random split of the original dataset. In other words, a holdout sets should necessarily independently and identically distributed with respect to the original dataset.
>
> As such, the holdout dataset and the public dataset do test the same things and a model that has not been contaminated by the public data should score similarly on the two dataset.
>
> Then our challenge is to show that one can construct such a holdout dataset retroactively.
>
> > "...the authors only validate their method's effectiveness on TruthfulQA with certain evaluation types and categories."
>
> We agree that it would have been better to verify our work on more datasets, which is why we will be including analysis on the other categories of TruthfulQA. All of these categories were initially developed independently to pass our tests, and were then iterated further to pass the tests when taken as a whole.
>
> Unfortunately, this won't include the *adversarial* type of question within the TruthfulQA evaluation, as our analysis indicates that many of its properties were significantly different from the *non-adversarial* set. E.g. we see that some GPT-3 models shows an accuracy difference between the adversarial and non-adversarial portion by up to 40 % -- which makes sense considering how it was constructed.
>
> As a result, creating a retro-holdout for the *adversarial* type would require use to utilize the same methodology for adversarial filtering of the original TruthfulQA *adversarial* questions. Unfortunately, OpenAI has discontinued access to the model that was used for this purpose.
>
> --
>
> Lastly, we want to clarify that we see our primary contribution is the demonstration that evaluation gaming, such as the data leakage which has been well documented in other works, is having a quantifiably significant impact on reported benchmark scores.

---

> > ### Author Response · Authors · 2024-08-27
> >
> > Dear Reviewer 1HP3,
> >
> > As the rebuttal period is approaching its conclusion, we wanted to follow up on our previous response to your review. We appreciate the time and effort you have invested in evaluating our work and hope our rebuttal addressed your main concerns.
> >
> > We would like to reiterate that while the mentioned limitation about real-world hold-out datasets is apt, we do not make use of these. Instead, our method relies on the more traditional definition where the holdout dataset is an independent and identically distributed (i.i.d.) split of the original dataset. This means that our retro-holdout dataset is designed to test the same properties as the original dataset and the gap in performance can be attributed to data being public.
> >
> > Additionally, while we recognize the importance of validating our method on more benchmarks, we focused on TruthfulQA due to its unique position in the timeline of LLM development.
> >
> > We hope these clarifications help to address the concerns you raised. If there are any remaining questions or uncertainties, we would greatly appreciate the opportunity to discuss them further before the review process concludes. If our responses have satisfactorily addressed your concerns, we kindly invite you to reconsider your initial score.
> >
> > Thank you again for your time and consideration.

---

> > ### Comment · Reviewer_1HP3 · 2024-08-31
> > **Acknowledge of the clarification of the rebuttal**
> >
> > Dear authors,
> > Thank you for clarifying. Your paper tackles a critical problem and helps us more fairly understand the models on public benchmarks. However, I think more experiments and different datasets are needed to make your work solid for the acceptance.

---

> > > ### Author Response · Authors · 2024-08-31
> > > **Acknowledgment and question on datasets**
> > >
> > > Kind reviewer,
> > >
> > > Thank you for all your feedback and for recognizing the significance of our work. We genuinely appreciate the time and effort you've put into reviewing our paper.
> > >
> > > For our benefit, could you clarify what exactly you'd expect regarding the inclusion of different datasets? Also, considering our primary goal was to show there's a real, measurable gap between actual performance and public dataset scores, why wouldn’t it be sufficient to demonstrate this on a key dataset like TruthfulQA? Should the focus not be more on ruling out other possible explanations for that gap rather than spreading the analysis across multiple datasets?
> > >
> > > Your input here would be really helpful.
> > >
> > > Thanks again for your review and insights.

---

### Official Review · Reviewer_hZPp · 2024-07-31

**Rating:** 5
**Confidence:** 3
**Correctness:** Yes
**Clarity:** Yes

**Review:**

Please refer to the strengths and Opportunities For Improvement

**Strengths:**

1.the most important contribution of this work is to quantify the inflation of benchmark performance for existing LLMs. It is a very useful topic for the research community, and should draw attention from other researchers.

2.the experimental results have well verified the motivation.

**Additional Feedback:**

No

**Documentation:**

Yes

**Ethics:**

No ethics issues

**Limitations:**

Yes

**Opportunities For Improvement:**

1.the experiments are not solid enough for supporting the conclusion. Actually, expect TruthfulQA, MMLU and CEval have been widely used for LLM comparison, which are much more proper benchmarks for investigation. In addition, TruthfulQA is not the best choice, as its focused human alignment capability has been a vital important topic in LLM training. Thus, it is hard to say if the post-training for human alignment objective has affected the inflation evaluation.

2.the motivation of the approach design is not clear. It seems like an adversarial method for evaluating the robustness of LLMs, but not the benchmark inflation. I suggest authors to discuss it and reorganize the whole paper.

**Relation To Prior Work:**

Yes

**Summary And Contributions:**

This paper shows significant discrepancies between benchmark performances and real-world capabilities of LLMs, underscoring the need for robust and reliable evaluation methodologies. A new methodology for constructing retro-holdout datasets is proposed in this work, and also indicates that for TruthfulQA. The results also prove that the above issue exists in the commonly-used benchmarks.

---

> ### Author Rebuttal · Authors · 2024-08-19
>
> Reviewer hZPp, we thank you for your review, and address your points below:
>
> > the motivation of the approach design is not clear. It seems like an adversarial method for evaluating robustness of LLMs, but not the benchmark inflation.
>
> We would like to emphasize that our method is not adversarial - no model has been involved in the construction of the retro holdout dataset, and the same retro holdout is used to evaluate every model.
>
> Rather, the approach is based on the concept of holdout datasets in machine learning. Holdout datasets are i.i.d. splits of full datasets which models have not been exposed to. That is, if there is no data leakage, models should perform the same on public test data and holdout test data, up to some statistical bound. If this is not the case, then we can conclude that the difference is due to some direct or indirect data leakage.
>
> Extending this to todays LLM benchmarks, if we had access to a private split of a public benchmark, we could compare model scores on both of these datasets to determine if benchmark scores are higher than they should be on the public versions. In this case, because we have controlled for all variables except for the publicity of the dataset, we would know that any significant difference would have to be a function of the public sets availability.
>
> > Actually, expect TruthfulQA, MMLU and CEval have been widely used for LLM comparison, which are much more proper benchmarks for investigation.
>
> We believe that TruthfulQA has been used extensively for model evaluation. Until recently, it was one of the benchmarks on the HuggingFace OpenLLM Leaderboard, which was frequently used for comparison within the last 2 years. As an additional example, Mistral used TruthfulQA as one of the benchmarks in it's release paper.
>
> There is another rather important reason, however. In order to prove that our method works, we need a dataset which was released between the first strong LLMs and the modern models. MMLU came out before GPT-3 while TruthfulQA conveniently was released after the first GPT-3 versions but well before the modern versions. This so that we can show that for models that were made before the release of the public dataset, they perform as well on the public data as our retroactively constructed dataset.
>
> > In addition, TruthfulQA is not the best choice, as its focused human alignment capability has been a vital important topic in LLM training Thus, it is hard to say if the post-training for human alignment objective has affected the inflation evaluation
>
> We agree that post-training processes for human alignment could have an effect on the inflation evaluation, and this is intentional. If post-training changes performance on the public dataset, but not a private (IID) split of that dataset, that would provide evidence that the reported scores are not representative of uncontaminated performance.
>
> You are right that we do not answer whether this inflation stems from the model's original pretraining or through its alignment tuning. However, we do not believe this actually matters. Whether through one effect or the other, we answer whether the models demonstrate benchmark inflation.
>
> > the experiments are not solid enough for supporting the conclusion
>
> We want to clarify that we see our most significant contribution as the demonstration that evaluation gaming, such as the data leakage which has been well documented in other works, is having a quantifiably significant impact on reported benchmark scores.
>
> To do so, we believe it is enough to demonstrate it conclusively on one dataset and that TruthfulQA enables that.
>
> We agree that it would be valuable to demonstrate that scores are inflated on other public datasets, but also believe that this already shows that the method works and that there is significant score inflation.

---

> > ### Author Response · Authors · 2024-08-27
> >
> > Reviewer hZPp,
> >
> > As we near the end of the rebuttal period, we wanted to verify our responses address all of your concerns with our submission. If there are remaining doubts beyond what’s discussed in our rebuttals, we want to ensure that we still have time to resolve them.
> >
> > We believe there may have been a misunderstanding regarding some key aspects of our work, and would like to clarify that the methodology is not adversarial - no model was involved in the construction of the retro holdout dataset. Our approach is based on the well-established concept of holdout datasets in machine learning, and we believe this distinction is crucial to accurately evaluating the contribution of our work.
> >
> > Additionally, while we agree that benchmarks like MMLU and CEval are important, we selected TruthfulQA for its specific relevance to our study’s goals. The timeline of TruthfulQA's release in relation to model development makes it a suitable choice for demonstrating the potential inflation in benchmark performance, which is a critical aspect of our findings.
> >
> > We kindly ask you to reconsider your evaluation in light of these clarifications. If any questions or uncertainties remain, we would greatly appreciate the opportunity to discuss them further before the review process concludes.
> >
> > We would be happy to respond to any further concerns or questions that you might have.

---

> > > ### Comment · Reviewer_hZPp · 2024-08-30
> > > **Response to Authors**
> > >
> > > Thank you for your clarification, and part of my concern has been addressed. Actually, this paper still needs more experiments and results on comprehensive evaluation benchmarks to make it more solid. So I keep my score.

---

> > > > ### Author Response · Authors · 2024-08-31
> > > > **Point on datasets**
> > > >
> > > > Thank you for your feedback and for recognizing the importance of our work. We appreciate the time you've put into your review.
> > > >
> > > > To clarify, our main goal was to show a measurable gap between real-world LLM performance and public dataset scores, with a focus on TruthfulQA due to its relevance and timing. We believe that demonstrating this gap on a single, carefully chosen dataset is sufficient to substantiate our claims and introduce our new methodology.
> > > >
> > > > It's important to emphasize that our approach is not adversarial; the dataset applies generally to all models, and intent tuning shouldn't be a factor. We also believe our statistical tests to ensure the retro-holdout dataset is indistinguishable are well-founded.
> > > >
> > > > While expanding to more benchmarks could further validate our findings, we believe our in-depth analysis of TruthfulQA already makes a strong case for benchmark inflation, addressing a broader issue in LLM evaluation.
> > > >
> > > > We're open to exploring more datasets in future work, but we believe our study provides a solid foundation for advancing the understanding of LLM evaluation.
> > > >
> > > > Thanks again for your insights.

---

### Decision · Program_Chairs · 2024-09-26

**Decision:**

Reject

**Comment:**

This paper proposed a new Retro-TruthfulQA dataset, based on the existing TruthfulQA dataset, to measure the inflation of evaluation due to data contamination, i.e., the evaluation dataset is publicly available. While reviewers acknowledge the importance of the problem, and the interesting observations, they are also concerned about the choice of the problem set, the generalizability of the procedure, and the comprehensiveness of the benchmark. This paper has some potential, but given the bar called out in meta review instructions, it would be difficult  to recommend acceptance.